# A fractional perspective on the transmission dynamics of a parasitic infection, considering the impact of both strong and weak immunity

Tao-Qian Tang[1,2,3,4,5☯], Rashid Jan[6☯], Zahir Shah[7☯]*, Narcisa Vrinceanu[8☯]*, Ciprian Tanasescu[9,10☯], Asif Jan[11☯]

1 Department of Internal Medicine, E-Da Hospital, I-Shou University, Kaohsiung, Taiwan, 2 School of Medicine, College of Medicine, I-Shou University, Kaohsiung, Taiwan, 3 International Intercollegiate Ph.D. Program, National Tsing Hua University, Hsinchu, Taiwan, 4 Department of Family and Community Medicine, E-Da Hospital, I-Shou University, Kaohsiung, Taiwan, 5 Department of Engineering and System Science, National Tsing Hua University, Hsinchu, Taiwan, 6 Department of Civil Engineering, College of Engineering, Institute of Energy Infrastructure (IEI), Universiti Tenaga Nasional (UNITEN), Putrajaya Campus, Kajang, Selangor, Malaysia, 7 Department of Mathematical Sciences, University of Lakki Marwat, Lakki Marwat, KPK, Pakistan, 8 Department of Industrial Machines and Equipments, Faculty of Engineering, "Lucian Blaga" University of Sibiu, Sibiu, Romania, 9 Preclin Dept, Fac Med, Lucian Blaga Univ Sibiu, Sibiu, Romania, 10 Dept Surg, Romania Sibiu Cty Clin Emergency Hosp, Sibiu, Romania, 11 Department of Pathogenic Microbiology & Immunology, School of Basic Medical Sciences, Xi'an Jiaotong University Health Science Center, Xi'an, China

☯ All these authors are contributed equally to this work.
* zahir@ulm.edu.pk (ZS); vrinceanu.narcisai@ulbsibiu.ro (NV)

**Data Availability Statement:** All relevant data are within the paper.

## Abstract

Infectious disease cryptosporidiosis is caused by the cryptosporidium parasite, a type of parasitic organism. It is spread through the ingestion of contaminated water, food, or fecal matter from infected animals or humans. The control becomes difficult because the parasite may remain in the environment for a long period. In this work, we constructed an epidemic model for the infection of cryptosporidiosis in a fractional framework with strong and weak immunity concepts. In our analysis, we utilize the well-known next-generation matrix technique to evaluate the reproduction number of the recommended model, indicated by $\mathfrak{R}_0$. As $\mathfrak{R}_0 < 1$, our results show that the disease-free steady-state is locally asymptotically stable; in other cases, it becomes unstable. Our emphasis is on the dynamical behavior and the qualitative analysis of cryptosporidiosis. Moreover, the fixed point theorem of Schaefer and Banach has been utilized to investigate the existence and uniqueness of the solution. We identify suitable conditions for the Ulam-Hyers stability of the proposed model of the parasitic infection. The impact of the determinants on the sickness caused by cryptosporidiosis is highlighted by the examination of the solution pathways using a novel numerical technique. Numerical investigation is conducted on the solution pathways of the system while varying various input factors. Policymakers and health officials are informed of the crucial factors pertaining to the infection system to aid in its control.

**Funding:** The funders had no role in study design, data collection and analysis, decision to publish, or preparation of the manuscript.

**Competing interests:** The authors have declared that no competing interests exist.

# 1 Introduction

Cryptosporidiosis is a type of enteric disease that occurs when an individual is infected by a parasite belonging to the cryptosporidium genus. It is a common waterborne disease and is responsible for many other such infections around the globe [1, 2]. Protozoal infections, particularly those caused by cryptosporidium and giardia produce over 58 million cases of diarrhea in children every year. The World Health Organization placed its attention on waterborne pathogens, including cryptosporidium and giardia [3, 4]. Although the infections of cryptosporidium are generally self-limiting and cause acute gastroenteritis in people with healthy immune systems, they can lead to chronic and potentially fatal diarrhea in people with compromised immune systems. Neonates are particularly vulnerable to infection as they can become infected with the parasite's oocysts by ingesting small amounts. Diarrheal diseases, which are often caused by contaminated water and poor hygiene, cause approximately 1.6 million deaths annually, and mostly under the age of five children are affected [5]. Up to 20% of all occurrences of diarrhea in children in underdeveloped nations are caused by Cryptosporidium, which can be fatal for HIV-positive individuals [6]. Cryptosporidiosis is most common among people living in urban and rural areas where the risk of disease transmission and spread is high [7, 8].

In 1982, the CDC published a report on patients infected with HIV who experienced diarrhea caused by cryptosporidium, highlighting the medical importance of the parasite in humans. In 1993, global interest in cryptosporidium as a public health issue grew after a waterborne outbreak in Milwaukee [9–11]. The number of waterborne cryptosporidium infections [12] doubled in the USA between 2014 and 2016, with approximately 748,000 cases of humans per year. In developing countries, inadequate sanitation of water and food elevates the likelihood of cryptosporidiosis, with the greatest impact observed among children under the age of five [13, 14]. Even in unfavorable circumstances, the cryptosporidium oocysts may last months without a host and continue to spread disease [15–17]. Calf diarrhea complex is mainly caused by different infections in calves with enterotoxin Escherichia coli, cryptosporidium, rotaviruses, and coronavirus [18]. There are currently no effective treatments for cryptosporidiosis [19, 20], although halofuginone and nitazoxanide are authorized drugs for humans and animals, their effectiveness is not guaranteed [21–23]. However, supportive measures can help manage the symptoms and reduce the severity of the infection.

The examination and evaluation of the dynamics of infectious disease transmission is significantly supported by mathematical models, which also aid in the development of efficient control measures. Some deterministic models have been created and scrutinized to comprehend the transmission dynamics of cryptosporidiosis, including [24–26]. A study in [25] conducted an analysis of optimal control measures for cryptosporidiosis in humans, whereas [24, 26] focused on investigating the dynamics of co-infections involving cryptosporidiosis and either trypanosomiasis and HIV-AIDS. There has been a lack of extensive research on the dynamics of cryptosporidiosis utilizing non-integer derivatives. Therefore, further investigation and study are necessary to fully comprehend the disease, as indicated by existing literature. Consequently, we present a comprehensive analysis of the intricate mechanisms underlying cryptosporidiosis to obtain a good comprehension of the disease transmission.

This work is structured as: Section 2 provides an overview of the fundamental findings of fractional calculus. Section 3 details the construction of a compartmental model for cryptosporidiosis disease. Furthermore, Section 3 calculates both the disease-free equilibrium and endemic equilibrium, as well as the reproduction number $\mathfrak{R}_0$. In Section 4, we evaluate the existence and uniqueness of the recommended system, while in Section 5, we pinpoint the necessary conditions for Ulam-Hyers stability. In Section 6, a numerical approach is introduced

to solve the proposed fractional system and to explore the response of cryptosporidiosis disease to various parameters. Section 7 contains the conclusion of the article.

## 2 Theory of fractional-calculus

The application of fractional theory is quite extensive in various research fields of science and engineering [27, 28]. To explore the model, the theory and results of the Caputo operator have been introduced:

[29] Consider a function $w(t)$ in a way that $w(t) \in E^1([f, \iota], \mathbb{R})$, then the fractional integral is

$$I_{f^+}^f w(t) = \frac{1}{\vartheta(\Theta)} \int_0^t (t - \hbar)^{\Theta-1} w(\hbar) d\hbar, \quad \Theta \in (0, 1], \tag{1}$$

in which the order of the fractional operator is denoted by $\Theta$.

[29]. Let us assume $w(t)$ in such a way that $w(t) \in C^\theta[f, \iota]$, then the Caputo operator is defined as

$$^C D_t^\Theta w(t) = \frac{1}{\vartheta(\theta - \Theta)} \int_0^t (t - \hbar)^{-(\Theta+1-\theta)} w^\theta(\hbar) d\hbar, \tag{2}$$

where $0 < \Theta \leq 1$. [29]. Take the below-mentioned system

$$\begin{cases} ^C D_{0^+}^\Re w(t) = & l(t), \quad 0 \leq t \leq v, \quad \theta - 1 < \Re < \theta, \\ w(0) = l_0, \end{cases} \tag{3}$$

where $l(t) \in C([0, v])$. The system (3) has a solution of the form

$$w(t) = \sum_{\iota=0}^{\theta-1} y_\iota t^\iota, in \ which \ y_\iota \in \mathbb{R}, \quad \iota = 0, 1, \ldots, \theta - 1. \tag{4}$$

[30]. For the Caputo operator, we have the below Laplace transform

$$\pounds[^\epsilon D_{0^+}^\Theta w(t)] = \hbar^\Theta w(\hbar) - \sum_{k=0}^{\theta-1} \hbar^{-(k+1-\Theta)} w^k(0), \theta - 1 < \Theta < \theta. \tag{5}$$

Additionally, let

$$||w|| = \max_{t \in [0,v]} \{|w|, for \ all \ w \in \mathcal{Z}\}, \tag{6}$$

is a norm on $\mathcal{Z} = C([0, v])$ where $\mathcal{Z}$ is a Banach space.

[31]. Let $\mathcal{Z}$ be a Banach space such that $U : \mathcal{Z} \to \mathcal{Z}$ is compact and continuous. If

$$L = \{w \in \mathcal{Z} : w = \wp Y w, \wp \in (0, 1)\}, \tag{7}$$

is bounded. Then one can find a fixed point of $U$.

## 3 Evaluation of the dynamics

In this formulation, the total human population is grouped into six classes, such as $\mathcal{S}_1$ represent the susceptible individual with strong immunity, $\mathcal{S}_2$ represent the susceptible individual with weakened immunity, $\mathcal{I}_1$ represent individual infected with strong immunity, $\mathcal{I}_2$ represent individual infected with weakened immunity, $\mathcal{T}$ represent treated individuals and $\mathcal{R}$ represent the recovered individual. The symbol $\mathcal{E}$ represents the contaminated environment. The

susceptible class of individuals with strong immunity increases at the rate $\rho(1 - \xi)$ and $\varpi\omega\mathcal{R}$, where $\rho$ represent recruitment rate of humans, $\xi$ represent the proportion of individuals with weakened immunity, $\omega$ represent the ingestion rate of cryptosporidiosis disease and $\varpi$ indicate the immunity waning rate. The class decreases at the rate $\varsigma\mathcal{S}_1$ and $\mu\beta\mathcal{S}_1$, where $\varsigma$ is the natural death rate of humans, $\mu$ is the modification parameter of infection rate and the force of infection is represented by $\beta$. Thus the class of susceptible individuals with weakened immunity is given as

$$\frac{d\mathcal{S}_1}{dt} = \rho(1 - \xi) + \varpi\omega\mathcal{R} - \varsigma\mathcal{S}_1 - \mu\beta\mathcal{S}_1.$$

The terms $\xi\rho$ and $(1 - \omega)\varpi\mathcal{R}$ increase the population of the susceptible individual with weakened immunity and the terms $\varsigma\mathcal{S}_2$ and $\beta\mathcal{S}_2$ lowers the population of the susceptible individuals with weakened immunity. Therefore the class of susceptible individuals with weakened immunity is stated as

$$\frac{d\mathcal{S}_2}{dt} = \xi\rho + (1 - \omega)\varpi\mathcal{R} - \varsigma\mathcal{S}_2 - \beta\mathcal{S}_2.$$

The class of individuals infected with strong immunity increases at the rate $\mu\beta\mathcal{S}_1$ and the class decreases at the rate $(\sigma + \varsigma + \alpha)\mathcal{I}_1$, where $\sigma$ is the recovery rate from the disease and $\alpha$ represents the deaths occur due to cryptosporidiosis disease. Thus the class of individuals infected with strong immunity is

$$\frac{d\mathcal{I}_1}{dt} = \mu\beta\mathcal{S}_1 - (\sigma + \varsigma + \alpha)\mathcal{I}_1.$$

The class of infected hosts with weakened immunity increases at $\beta\mathcal{S}_2$ and decreases at $(\varrho + \varsigma + \alpha)\mathcal{I}_2$, where $\varrho$ represent the individual infected with weakened immunity treatment rate. Therefore the class of infected hosts with weakened immunity is given by

$$\frac{d\mathcal{I}_2}{dt} = \beta\mathcal{S}_2 - (\varrho + \varsigma + \alpha)\mathcal{I}_2.$$

The treated individuals grow at the rate $\varrho\mathcal{I}_2$ and lower at the rate $(\chi + \varsigma)\mathcal{T}$, where $\chi$ indicate the treated individuals with weakened immune system recovery rate from the disease. Thus the treated individual class is mathematically represented by

$$\frac{d\mathcal{T}}{dt} = \varrho\mathcal{I}_2 - (\chi + \varsigma)\mathcal{T}.$$

The recovered hosts grow at $\sigma\mathcal{I}_1$ and $\chi\mathcal{T}$, and decreases at the rate $(\varsigma + \varpi)\mathcal{R}$. Therefore the recovered individuals stated as

$$\frac{d\mathcal{R}}{dt} = \sigma\mathcal{I}_1 + \chi\mathcal{T} - (\varsigma + \varpi)\mathcal{R}.$$

The contaminated environment class increases at the rate $(\mathcal{I}_1 + \mathcal{I}_2)\gamma$ and decreases at the rate $\kappa\mathcal{E}$, where $\kappa$ represent the rate at which cryptos leaves the environment. Thus, the contaminated environment class is mathematically given as

$$\frac{d\mathcal{E}}{dt} = (\mathcal{I}_1 + \mathcal{I}_2)\gamma - \kappa\mathcal{E}.$$

Now, writing all the above in the form of a system, we have

$$
\begin{cases}
\frac{d\mathcal{S}_1}{dt} &=& \rho(1-\xi) + \varpi\omega\mathcal{R} - \varsigma\mathcal{S}_1 - \mu\beta\mathcal{S}_1, \\
\frac{d\mathcal{S}_2}{dt} &=& \xi\rho + (1-\omega)\varpi\mathcal{R} - \varsigma\mathcal{S}_2 - \beta\mathcal{S}_2, \\
\frac{d\mathcal{I}_1}{dt} &=& \mu\beta\mathcal{S}_1 - (\sigma + \varsigma + \alpha)\mathcal{I}_1, \\
\frac{d\mathcal{I}_2}{dt} &=& \beta\mathcal{S}_2 - (\varrho + \varsigma + \alpha)\mathcal{I}_2, \\
\frac{d\mathcal{T}}{dt} &=& \varrho\mathcal{I}_2 - (\chi + \varsigma)\mathcal{T}, \\
\frac{d\mathcal{R}}{dt} &=& \sigma\mathcal{I}_1 + \chi\mathcal{T} - (\varsigma + \varpi)\mathcal{R}, \\
\frac{d\mathcal{E}}{dt} &=& (\mathcal{I}_1 + \mathcal{I}_2)\gamma - \kappa\mathcal{E},
\end{cases}
\tag{8}
$$

where

$$
\beta = \varphi\tau(\mathcal{I}_1 + \mathcal{I}_2) + \eta\mathcal{E}.
$$

The strength of the host is denoted by $\mathcal{N}$ and can be written as

$$
\mathcal{N} = \mathcal{S}_1 + \mathcal{S}_2 + \mathcal{I}_1 + \mathcal{I}_2 + \mathcal{T} + \mathcal{R},
$$

with the following

$$
\mathcal{S}_1 \geq 0, \mathcal{S}_2 \geq 0, \mathcal{I}_1 \geq 0, \mathcal{I}_2 \geq 0, \mathcal{T} \geq 0, \mathcal{R} \geq 0, \mathcal{E} \geq 0.
$$

Fractional calculus is capable of providing adequate models for numerous biological and engineering issues. Fractional epidemic models provide a more accurate representation of real-world epidemics compared to classical models. Most biological processes possess information about their past and are nonlocal which can not be represented accurately through ordinary models. On the other hand, fractional derivatives possess hereditary prosperity and accurately illustrate the nonlocal behavior of epidemic models. Moreover, these models provide an extra parameter during real data fitting. Therefore, we represent our model (8) of parasitic infection as

$$
\begin{cases}
{}^{LC}_{0}D_t^{\Re}\mathcal{S}_1 &=& \rho(1-\xi) + \varpi\omega\mathcal{R} - \varsigma\mathcal{S}_1 - \mu\beta\mathcal{S}_1, \\
{}^{LC}_{0}D_t^{\Re}\mathcal{S}_2 &=& \xi\rho + (1-\omega)\varpi\mathcal{R} - \varsigma\mathcal{S}_2 - \beta\mathcal{S}_2, \\
{}^{LC}_{0}D_t^{\Re}\mathcal{I}_1 &=& \mu\beta\mathcal{S}_1 - (\sigma + \varsigma + \alpha)\mathcal{I}_1, \\
{}^{LC}_{0}D_t^{\Re}\mathcal{I}_2 &=& \beta\mathcal{S}_2 - (\varrho + \varsigma + \alpha)\mathcal{I}_2, \\
{}^{LC}_{0}D_t^{\Re}\mathcal{T} &=& \varrho\mathcal{I}_2 - (\chi + \varsigma)\mathcal{T}, \\
{}^{LC}_{0}D_t^{\Re}\mathcal{R} &=& \sigma\mathcal{I}_1 + \chi\mathcal{T} - (\varsigma + \varpi)\mathcal{R}, \\
{}^{LC}_{0}D_t^{\Re}\mathcal{E} &=& (\mathcal{I}_1 + \mathcal{I}_2)\gamma - \kappa\mathcal{E},
\end{cases}
\tag{9}
$$

where the symbol $\Re$ is used to represent the fractional order of the derivative, while ${}^{LC}_{0}D_t^{\Re}$ represents the Liouville-Caputo derivative. The parameter values of the model are described in Table 1.

**Table 1. Illustration of the input parameters of the proposed model with interpretation.**

| parameter | interpretation |
|---|---|
| $\omega$ | Cryptosporidiosis disease ingestion rate. |
| $\xi$ | The population of humans with weakened immunity. |
| $\mu$ | The infection rate modification parameter. |
| $\rho$ | Recruitment rate of individuals. |
| $\tau$ | Contact rate of individuals. |
| $\beta$ | The force of infection. |
| $\varphi$ | Transmission probability of disease. |
| $\sigma$ | Recovery rate from disease. |
| $\varrho$ | The infected individuals with weakened immune system treatment rate from the disease. |
| $\chi$ | The treated individuals with weakened immune system recovery rate from the disease. |
| $\gamma$ | Each cryptosporidiosis infected individual average contribution to the environment. |
| $\varpi$ | Immune waning rate. |
| $\alpha$ | Deaths occur due to cryptosporidiosis. |
| $\kappa$ | The cryptos leaving rate form the environment. |

## 3.1 Analysis of the model

Here, we will examine the suggested fractional model of cryptosporidiosis. In order to achieve the infection free-equilibrium, model (9) in steady state can be written as

$$
\begin{cases}
0 &= \rho(1-\xi) + \varpi\omega\mathcal{R} - \varsigma\mathcal{S}_1 - \mu\beta\mathcal{S}_1, \\
0 &= \xi\rho + (1-\omega)\varpi\mathcal{R} - \varsigma\mathcal{S}_2 - \beta\mathcal{S}_2, \\
0 &= \mu\beta\mathcal{S}_1 - (\sigma+\varsigma+\alpha)\mathcal{I}_1, \\
0 &= \beta\mathcal{S}_2 - (\varrho+\varsigma+\alpha)\mathcal{I}_2, \\
0 &= \varrho\mathcal{I}_2 - (\chi+\varsigma)\mathcal{T}, \\
0 &= \sigma\mathcal{I}_1 + \chi\mathcal{T} - (\varsigma+\varpi)\mathcal{R}, \\
0 &= (\mathcal{I}_1+\mathcal{I}_2)\gamma - \kappa\mathcal{E}.
\end{cases}
\tag{10}
$$

When there is no infection present, the equilibrium point obtained from the system (10) is known as the disease-free equilibrium. If we assume that all the infected classes have a value of zero ($\mathcal{I}_1 = \mathcal{I}_2 = 0$), we can derive the following outcome;

$$
\mathbf{E_0} = (\mathcal{S}_1, \mathcal{S}_2, \mathcal{I}_1, \mathcal{I}_2, \mathcal{T}, \mathcal{R}, \mathcal{E}) = \left( \frac{\rho(1-\xi)}{\varsigma}, \frac{\rho\xi}{\varsigma}, 0, 0, 0, 0, 0 \right).
$$

Further, we illustrate the local asymptotic stability result for the infection-free equilibrium of the recommended model of parasitic infection. If $\mathfrak{R}_0 < 1$, then the equilibrium $\mathbf{E_0}$ of the system is locally asymptotically stable and otherwise unstable. By taking the Jacobian matrix of

system (9) at DFE, we achieve that

$$
\mathfrak{J}_1 = \begin{bmatrix}
-\varsigma & 0 & \frac{-\mu\varphi\tau\rho(1-\xi)}{\varsigma} & \frac{-\mu\varphi\tau\rho(1-\xi)}{\varsigma} & 0 & \varpi\omega & \frac{-\mu\eta\rho(1-\xi)}{\varsigma} \\
0 & -\varsigma & \frac{-\varphi\tau\rho\xi}{\varsigma} & \frac{-\varphi\tau\rho\xi}{\varsigma} & 0 & (1-\omega)\varpi & \frac{-\eta\rho\xi}{\varsigma} \\
0 & 0 & \mathcal{L} & \frac{\mu\varphi\tau\rho(1-\xi)}{\varsigma} & 0 & 0 & \frac{\mu\eta\rho(1-\xi)}{\varsigma} \\
0 & 0 & \frac{\varphi\tau\rho\xi}{\varsigma} & \frac{\varphi\tau\rho\xi}{\varsigma} - (\varrho + \varsigma + \alpha) & 0 & 0 & \frac{\eta\rho\xi}{\varsigma} \\
0 & 0 & 0 & \varrho & -(\chi + \varsigma) & 0 & 0 \\
0 & 0 & \sigma & 0 & \chi & -(\varsigma + \varpi) & 0 \\
0 & 0 & \gamma & \gamma & 0 & 0 & -\kappa
\end{bmatrix},
$$

where

$\mathcal{L} = \frac{\mu\varphi\tau\rho(1-\xi)}{\varsigma} - (\sigma + \varsigma + \alpha).$

Clearly, from the matrix, we observe that in the first two columns, the non-diagonal elements are zero, by further evaluation we get two negative eigenvalues that are $-\varsigma$ and $-\varsigma$. In order to obtain the remaining eigenvalues, the above matrix can be reduced into the following matrix by removing the first two rows and columns

$$
\mathfrak{J}_2 = \begin{bmatrix}
\mathcal{L} & \frac{\mu\varphi\tau\rho(1-\xi)}{\varsigma} & 0 & 0 & \frac{\mu\eta\rho(1-\xi)}{\varsigma} \\
\frac{\varphi\tau\rho\xi}{\varsigma} & \frac{\varphi\tau\rho\xi}{\varsigma} - (\varrho + \varsigma + \alpha) & 0 & 0 & \frac{\eta\rho\xi}{\varsigma} \\
0 & \varrho & -(\chi + \varsigma) & 0 & 0 \\
\sigma & 0 & \chi & -(\varsigma + \varpi) & 0 \\
\gamma & \gamma & 0 & 0 & -\kappa
\end{bmatrix}.
$$

Let the eigenvalues be $\mathcal{V}_i, \ i = 1, 2, 3, 4, 5$, then we have

$$
Det(\mathfrak{J}_2 - \mathcal{V}I) = \begin{bmatrix}
\mathcal{L} - \mathcal{V} & \frac{\mu\varphi\tau\rho(1-\xi)}{\varsigma} & 0 & 0 & \frac{\mu\eta\rho(1-\xi)}{\varsigma} \\
\frac{\varphi\tau\rho\xi}{\varsigma} & \frac{\varphi\tau\rho\xi}{\varsigma} - (\varrho + \varsigma + \alpha) - \mathcal{V} & 0 & 0 & \frac{\eta\rho\xi}{\varsigma} \\
0 & \varrho & -(\chi + \varsigma) - \mathcal{V} & 0 & 0 \\
\sigma & 0 & \chi & -(\varsigma + \varpi) - \mathcal{V} & 0 \\
\gamma & \gamma & 0 & 0 & -\kappa - \mathcal{V}
\end{bmatrix} = 0.
$$

The eigenvalues of the above will be the root of the following

$$
\mathfrak{N}^5 + \mathfrak{N}^4\mathfrak{F}_1 + \mathfrak{N}^3\mathfrak{F}_2 + \mathfrak{N}^2\mathfrak{F}_3 + \mathfrak{N}^1\mathfrak{F}_4 + \mathfrak{F}_5 = 0,
$$

where

$\mathfrak{F}_1 = \varsigma(\sigma + \varrho + \chi + 4\varsigma + \kappa + \varpi + 2\alpha) + (1 - \xi)\mu\varphi\tau\rho - \xi\varphi\tau\rho,$
$\mathfrak{F}_2 = (F_{21} + F_{22} + F_{23} + \varsigma(F_{24} + F_{25})),$
$\mathfrak{F}_3 = (F_{31} + \varsigma(F_{32} + F_{33}) + \alpha F_{34} + F_{35} + F_{36}),$
$\mathfrak{F}_4 = F_{41} + (\xi - 1)\mu\varphi\tau\rho F_{42} + \varsigma(F_{43} + \sigma F_{44}) + (F_{45} + \varsigma F_{46})\alpha + F_{47},$
$\mathfrak{F}_5 = \varsigma\kappa(\varsigma + \chi)(\varsigma + \varpi)(\sigma + \varsigma + \alpha)(\varrho + \varsigma + \alpha)(1 - \mathfrak{R}_0).$
with
$F_{21} = (\xi\eta + \mu\eta - \xi\mu\eta)(-\gamma\rho),$
$F_{22} = -\xi\varphi\tau\rho(\sigma + \chi + 3\varsigma + \kappa + \varpi + \alpha),$
$F_{23} = (\xi - 1)\mu\varphi\tau\rho(\varrho + \chi + 3\varsigma + \kappa + \varpi + \alpha),$
$F_{24} = 3\chi\varsigma + 6\varsigma^2 + \chi\kappa + 4\varsigma\kappa + \chi\varpi + 3\varsigma\varpi + \kappa\varpi + 2(\chi + 3\varsigma + \kappa + \varpi)\alpha + \alpha^2,$

$F_{25} = \sigma\varrho + (\sigma + \varrho)(\chi + 3\varsigma + \kappa + \varpi + \alpha)$,

$F_{31} = (\xi - 1)\mu\eta\gamma\rho(\varrho + \chi + 3\varsigma + \varpi) - \xi\eta\gamma\rho(\sigma + \chi + 3\varsigma + \varpi)$,

$F_{32} = \varsigma(3\chi(\varsigma + \kappa) + 2\varsigma(2\varsigma + 3\kappa)) + (3\varsigma(\varsigma + \kappa) + \chi(2\varsigma + \kappa))\varpi$,

$F_{33} = \varrho(3\varsigma(\varsigma + \kappa) + (2\varsigma + \kappa)\varpi + \chi(2\varsigma + \kappa + \varpi)) + \sigma(3\varsigma(\varsigma + \kappa) + (2\varsigma + \kappa)\varpi$
$+ \chi(2\varsigma + \kappa + \varpi) + \varrho(\chi + 2\varsigma + \kappa + \varpi))$,

$F_{34} = \varsigma(\sigma(\chi + 2\varsigma + \kappa + \varpi) + \varrho(\chi + 2\varsigma + \kappa + \varpi) + 2(3\varsigma(\varsigma + \kappa) + (2\varsigma + \kappa)\varpi$
$+ \chi(2\varsigma + \kappa + \varpi))) - (\xi\eta + \mu\eta - \xi\mu\eta)\gamma\rho$,

$F_{35} = (-\xi\varphi\tau\rho)(3\varsigma^2 + 3\varsigma\kappa + 2\varsigma\varpi + \kappa\varpi + \sigma(\chi + 2\varsigma + \kappa + \varpi) + (2\varsigma + \kappa + \varpi)$
$\alpha + \chi(2\varsigma + \kappa + \varpi + \alpha)) + \varsigma(\chi + 2\varsigma + \kappa + \varpi)\varsigma^2$,

$F_{36} = (\xi - 1)\mu\varphi\tau\rho(3\varsigma^2 + 3\varsigma\kappa + 2\varsigma\varpi + \kappa\varpi + \varrho(\chi + 2\varsigma + \kappa + \varpi) + (2\varsigma$
$+ \kappa + \varpi)\alpha + \chi(2\varsigma + \kappa + \varpi + \alpha))$,

$F_{41} = (-\xi\varphi\tau\rho)(\chi\varsigma(\varsigma + 2\kappa) + \chi(\varsigma + \kappa)\varpi + \varsigma(\varsigma^2 + 3\varsigma\kappa + \varsigma\varpi + 2\kappa\varpi) + \sigma(\varsigma(\varsigma + 2\kappa)$
$+ (\varsigma + \kappa)\varpi + \chi(\varsigma + \kappa + \varpi)))$,

$F_{42} = \chi\varsigma(\varsigma + 2\kappa) + \varsigma(\varsigma + \kappa)\varpi + \varsigma(\varsigma^2 + 3\varsigma\kappa + \varsigma\varpi + 2\kappa\varpi) + \varrho(\varsigma(\varsigma + 2\kappa) +$
$(\varsigma + \kappa)\varpi + \chi(\varsigma + \kappa + \varpi))$,

$F_{43} = \varsigma(\varsigma(\varrho + \varsigma)(\chi + \varsigma) + (2\varrho\chi + 3(\varrho + \chi)\varsigma + 4\varsigma^2)\kappa) + (\varsigma(\varrho + \varsigma)(\chi + \varsigma) + (\varrho\chi +$
$2(\varrho + \chi)\varsigma + 3\varsigma^2)\kappa)\varpi$,

$F_{44} = (\chi\varsigma(\varsigma + 2\kappa) + \chi(\varsigma + \kappa)\varpi + \varsigma(\varsigma^2 + 3\varsigma\kappa + \varsigma\varpi + 2\kappa\varpi) + \varrho(\varsigma(\varsigma + 2\kappa) + (\varsigma + \kappa)$
$\varpi + \chi(\varsigma + \kappa + \varpi)))$,

$F_{45} = (\xi - 1)\mu\varphi\tau\rho(\varsigma(\varsigma + 2\kappa) + (\varsigma + \kappa)\varpi + \chi(\varsigma + \kappa + \varpi)) - \xi\varphi\tau\rho(\varsigma(\varsigma + 2\kappa) + (\varsigma +$
$\kappa)\varpi + \chi(\varsigma + \kappa + \varpi))$,

$F_{46} = 2\varsigma(\chi(\varsigma + 2\kappa) + \varsigma(\varsigma + 3\kappa)) + 2(\chi(\varsigma + \kappa) + \varsigma(\varsigma + 2\kappa))\varpi + \sigma(\varsigma(\varsigma + 2\kappa) + (\varsigma$
$+ \kappa)\varpi + \chi(\varsigma + \kappa + \varpi)) + \varrho(\varsigma(\varsigma + 2\kappa) + (\varsigma + \kappa)\varpi + \chi(\varsigma + \kappa + \varpi))$,

$F_{47} = \varsigma(\varsigma(\varsigma + 2\kappa) + (\varsigma + \kappa)\varpi + \chi(\varsigma + \kappa + \varpi))\alpha^2 - \xi\eta\gamma\rho(3\varsigma^2 + 2\varsigma\varpi + \sigma(\chi + 2\varsigma + \varpi) +$
$2\varsigma\alpha + \varpi\alpha + \chi(2\varsigma + \varpi + \alpha)) + (\xi - 1)\mu\eta\gamma\rho(3\varsigma^2 + 2\varsigma\varpi + \varrho(\chi + 2\varsigma + \varpi) + 2\varsigma\alpha + \varpi\alpha +$
$\chi(2\varsigma + \varpi + \alpha))$,

which shows that the DFE is locally asymptotically stable if $\mathfrak{R}_0 < 1$ and unstable in other cases. The field of epidemiology considers the reproduction number ($\mathfrak{R}_0$) as a valuable parameter [32]. It estimates the number of new infections due to a primary infection in a susceptible population. To calculate ($\mathfrak{R}_0$), we will utilize the method described in [33], which involves the two matrices **V** and **F**. To determine the reproduction parameter, we proceed as

$$\mathbf{F} = \begin{bmatrix} \mu\beta\mathcal{S}_1 \\ \beta\mathcal{S}_2 \\ 0 \\ 0 \end{bmatrix}, \text{ and } \mathbf{V} = \begin{bmatrix} (\sigma + \varsigma + \alpha)\mathcal{I}_1 \\ (\varrho + \varsigma + \alpha)\mathcal{I}_2 \\ \varrho\mathcal{I}_2 + (\chi + \varsigma)\mathcal{T} \\ -(\mathcal{I}_1 + \mathcal{I}_2)\gamma + \kappa\mathcal{E} \end{bmatrix}.$$

Taking the jacobian of **F** and **V** at $\mathbf{E_0}$, yields the following

$$f = \begin{bmatrix} \frac{\mu\varphi\tau\rho(1-\xi)}{\varsigma} & \frac{\mu\varphi\tau\rho(1-\xi)}{\varsigma} & 0 & \frac{\mu\eta\rho(1-\xi)}{\varsigma} \\ \frac{\varphi\tau\xi\rho}{\varsigma} & \frac{\varphi\tau\xi\rho}{\varsigma} & 0 & \frac{\eta\xi\rho}{\varsigma} \\ 0 & 0 & 0 & 0 \\ 0 & 0 & 0 & 0 \end{bmatrix}, \text{ and } v = \begin{bmatrix} \sigma + \varsigma + \alpha & 0 & 0 & 0 \\ 0 & \varrho + \varsigma + \alpha & 0 & 0 \\ 0 & -\varrho & \chi + \varsigma & 0 \\ 0 & 0 & 0 & \kappa \end{bmatrix}.$$

Now taking the inverse of matric $v$, we get

$$
v^{-1} = \begin{bmatrix} \frac{1}{\sigma+\varsigma+\alpha} & 0 & 0 & 0 \\ 0 & \frac{1}{\varrho+\varsigma+\alpha} & 0 & 0 \\ 0 & \frac{\varrho}{(\chi+\varsigma)(\varrho+\varsigma+\alpha)} & \frac{1}{\chi+\varsigma} & 0 \\ 0 & 0 & 0 & \frac{1}{\kappa} \end{bmatrix}. \tag{11}
$$

$$
fv^{-1} = \begin{bmatrix} \frac{(1-\xi)\mu\varphi\tau\rho}{\varsigma(\sigma+\varsigma+\alpha)} & \frac{(1-\xi)\mu\varphi\tau\rho}{\varsigma(\varrho+\varsigma+\alpha)} & 0 & \frac{(1-\xi)\mu\eta\rho}{\varsigma\kappa} \\ \frac{\varphi\tau\xi\rho}{\varsigma(\sigma+\varsigma+\alpha)} & \frac{\varphi\tau\xi\rho}{\varsigma(\varrho+\varsigma+\alpha)} & 0 & \frac{\eta\xi\rho}{\varsigma\kappa} \\ 0 & 0 & 0 & 0 \\ 0 & 0 & 0 & 0 \end{bmatrix}. \tag{12}
$$

From the above, we determine the spectral radius of $fv^{-1}$ which is the reproduction parameter and is

$$
\mathfrak{R}_0 = \frac{\mu\varphi\tau\varrho\varrho\kappa - \xi\mu\varphi\tau\varrho\varrho\kappa + \sigma\varphi\tau\xi\rho\kappa + \mu\varphi\tau\rho\varsigma\kappa - \xi\mu\varphi\tau\rho\varsigma\kappa + \varphi\tau\xi\rho\varsigma\kappa + \mu\varphi\tau\rho\kappa\alpha + \varphi\tau\xi\rho\kappa\alpha - \xi\mu\varphi\tau\rho\kappa\alpha}{\varsigma\kappa(\sigma+\varsigma+\alpha)(\varrho+\varsigma+\alpha)}.
$$

The basic reproduction number is an important concept in public health, as it helps researchers and policymakers to understand the potential impact of an infection and to design effective control policies to mitigate its spread.

In the upcoming step, we will interrogate the endemic equilibrium of the model. We proceed with the system (10) as

$$
\begin{cases}
\mathcal{S}_1^* &= \frac{(1-\xi)\rho(\varsigma+\varpi)(\chi+\varsigma)+\omega\varpi((\chi+\varsigma)\sigma\mathcal{I}_1^*+\chi\varrho\mathcal{I}_2^*)}{(\varsigma+\mu\beta)(\varsigma+\varpi)(\chi+\varsigma)}, \\[2mm]
\mathcal{S}_2^* &= \frac{\xi\rho(\varsigma+\varpi)(\chi+\varsigma)+(1-\omega)\varpi((\chi+\varsigma)\sigma\mathcal{I}_1^*+\chi\varrho\mathcal{I}_2^*)}{(\varsigma+\mu\beta)(\varsigma+\varpi)(\chi+\varsigma)}, \\[2mm]
\mathcal{I}_1^* &= \frac{\mu(\rho(1-\xi)(\varsigma+\varpi)(\chi+\varsigma)+\omega\varpi\chi\varrho\mathcal{I}_2^*)\beta}{(\chi+\varsigma)(\varsigma(\varsigma+\varpi)(\sigma+\varsigma+\alpha)+\mu((\varsigma+\varpi)(\varsigma+\alpha)+\sigma(\varsigma+\varpi+\omega\varpi)))\beta}, \\[2mm]
\mathcal{I}_2^* &= \frac{(\xi\rho(\varsigma+\varpi)(\chi+\varsigma)+(1-\omega)\varpi(\sigma(\chi+\varsigma)\mathcal{I}_1^*+\varrho\chi\mathcal{I}_2^*))\beta}{\varsigma(\chi+\varsigma)(\varsigma+\varpi)(\varrho+\varsigma+\alpha)+((\varsigma+\varpi)(\varsigma+\chi)(\varrho+\varsigma+\alpha)-(1-\omega)\varpi\chi\varrho)\beta}, \\[2mm]
\mathcal{T}^* &= \frac{\varrho\mathcal{I}_2^*}{\chi+\varsigma}, \\[2mm]
\mathcal{R}^* &= \frac{(\chi+\varsigma)\sigma\mathcal{I}_1^*+\chi\varrho\mathcal{I}_2^*}{(\chi+\varsigma)(\varpi+\varsigma)}, \\[2mm]
\mathcal{E}^* &= \frac{(\mathcal{I}_1^*+\mathcal{I}_2^*)\gamma}{\kappa}.
\end{cases} \tag{13}
$$

To obtain the endemic equilibrium stability, from the value of $\beta$, we have

$$
\beta^* - \varphi\tau(\mathcal{I}_1^* + \mathcal{I}_2^*) - \eta\mathcal{E}^* = 0 \tag{14}
$$

Putting Eqs (13) into (14), then we get that the endemic value of $\beta^*$ satisfies the below stated

cubic polynomial as

$$\mathfrak{M}_1 \beta^{*3} + \mathfrak{M}_2 \beta^{*2} + \mathfrak{M}_3 \beta^* = 0$$

$$\beta^*(\mathfrak{M}_1 \beta^{*2} + \mathfrak{M}_2 \beta^* + \mathfrak{M}_3) = 0 \tag{15}$$

$\beta^* = 0$, is a root of cubic Eq (15), which correspond to the disease-free equilibrium. To obtain the other roots we have to solve;

$$\mathfrak{M}_1 \beta^{*2} + \mathfrak{M}_2 \beta^* + \mathfrak{M}_3 = 0$$

where

$\mathfrak{M}_1 = \mu\kappa(\mathcal{A}_{11} + \chi(\mathcal{A}_{12} + \sigma\mathcal{A}_{13}))$,

$\mathfrak{M}_2 = (\varsigma + \varpi)(\mathcal{A}_{21} + \xi\mu\rho(\tau\varphi\rho + \eta\gamma)\mathcal{A}_{22} + \mu(\chi(\mathcal{A}_{23} + \mathcal{A}_{24} + \mathcal{A}_{25} + \mathcal{A}_{26}) + \kappa(\mathcal{A}_{27} + \alpha\mathcal{A}_{28} + \mathcal{A}_{29} + \varrho\mathcal{A}_{20})))$,

$\mathfrak{M}_3 = \varsigma^2\kappa(\chi + \varsigma)(\varsigma + \varpi)^2(\sigma + \varsigma + \alpha)(\varrho + \varsigma + \alpha)(1 - \mathfrak{R}_0)$,

with $\mathcal{A}_{11} = \varsigma(\varsigma + \varpi)(\varrho + \varsigma\alpha)(\sigma(\varsigma + \varpi - \omega\varpi)(\varsigma + \alpha))$,

$\mathcal{A}_{12} = (\varsigma + \varpi)(\varsigma + \alpha)(\varrho(\varsigma + \omega\varpi) + (\varsigma + \varpi)(\varsigma + \alpha))$,

$\mathcal{A}_{13} = \varrho(\varsigma^2 + \varsigma\varpi + 2\omega\varpi_2 - 2\omega_2\varpi_2) + (\varsigma + \varpi)(\varsigma + \varpi - \omega\varpi)(\varsigma_n + \alpha)$,

$\mathcal{A}_{21} = \varsigma\kappa(\sigma + \varsigma + \alpha)(\varsigma(\varsigma + \varpi)(\varrho + \varsigma + \alpha) + \chi(\varrho(\varsigma + \omega\varpi) + (\varsigma + \varpi)(\varsigma + \alpha)))$,

$\mathcal{A}_{22} = (-\chi(\varrho\varsigma + \sigma(\varsigma - 2(-1 + \omega)\varpi) + 2(\varsigma + \varpi)(\varsigma + \alpha)) + \varsigma(\sigma(\varsigma - 2(-1 + \omega)\varpi) + (\varsigma + \varpi)(\varrho + 2(\varsigma + \alpha))))$,

$\mathcal{A}_{23} = \sigma\varsigma^3\kappa + \varsigma^4\kappa + \gamma\rho\varsigma^2\eta + \sigma\varsigma^2\kappa\varpi - \omega\sigma\varsigma^2\kappa\varpi + \varsigma^3\kappa\varpi + \sigma\gamma\rho\eta\varpi - \omega\sigma\gamma\rho\eta\varpi + \gamma\rho\varsigma\eta\varpi$,

$\mathcal{A}_{24} = ((2\varsigma^2\rho + \gamma\rho\eta)(\varsigma + \varpi) + \sigma\varsigma\kappa(\varsigma + \varpi - \omega\varpi))\alpha + \varsigma\kappa(\varsigma + \varpi)\varsigma^2$,

$\mathcal{A}_{25} = \varrho(\varsigma^3\kappa + \gamma\rho\varsigma\eta + \varsigma^2\kappa\varpi + \omega\gamma\rho\eta\varpi + \sigma\varsigma\kappa(\varsigma + \varpi - \omega\varpi) + \tau\varphi\rho\kappa(\varsigma + \omega\varpi) + \varsigma\kappa(\varsigma + \varpi)\alpha)$,

$\mathcal{A}_{26} = \tau\varphi\rho\kappa((1 - \omega)\sigma\varpi + (\varsigma + \varpi)(\varsigma + \alpha))$,

$\mathcal{A}_{27} = \sigma\varsigma^3\kappa + \varsigma^4\kappa + \gamma\rho\varsigma^2\eta + \sigma\varsigma^2\kappa\varpi - \omega\sigma\varsigma^2\kappa\varpi + \varsigma^3\kappa\varpi + \sigma\gamma\rho\eta\varpi - \omega\sigma\gamma\rho\eta\varpi + \gamma\rho\varsigma\eta\varpi$,

$\mathcal{A}_{28} = ((2\varsigma^2\kappa + \gamma\rho\eta)(\varsigma + \varpi) + \sigma\varsigma\kappa(\varsigma + \varpi - \omega\varpi))$,

$\mathcal{A}_{29} = \varsigma\kappa(\varsigma + \varpi)\alpha^2 + \tau\varphi\rho\kappa(1 - \omega)\sigma\varpi + (\varsigma + \varpi)(\varsigma + \alpha)$,

$\mathcal{A}_{20} = (\tau\varphi\rho\kappa(\varsigma + \varpi) + \sigma\varsigma\kappa(\varsigma + \varpi - \omega\varpi) + (\varsigma + \varpi)(\gamma\rho\eta + \varsigma\kappa(\varsigma + \alpha)))$.

Obviously, if $\mathfrak{R}_0 > 1$, then we get an unstable DFE which shows that there exists an endemic equilibrium that is locally asymptotically stable.

## 4 Theory of existence

Although some investigations have been conducted on qualitative fractional calculus theories, research in this area is generally limited [29, 34]. The need to find solutions for multiple models has prompted researchers to accelerate their work. To investigate the existence of solutions of models, researchers have explored [35, 36]. In this article, we will focus on the qualitative

analysis of the model (9) of parasitic infection. We proceed as follows

$$\begin{cases} \mathfrak{Y}_1(t, \mathcal{S}_1, \mathcal{S}_2, \mathcal{I}_1, \mathcal{I}_2, \mathcal{T}, \mathcal{R}, \mathcal{E}) & = & \rho(1-\xi) + \varpi\omega\mathcal{R} - \varsigma\mathcal{S}_1 - \mu\beta\mathcal{S}_1, \\ \mathfrak{Y}_2(t, \mathcal{S}_1, \mathcal{S}_2, \mathcal{I}_1, \mathcal{I}_2, \mathcal{T}, \mathcal{R}, \mathcal{E}) & = & \xi\rho + (1-\omega)\varpi\mathcal{R} - \varsigma\mathcal{S}_2 - \beta\mathcal{S}_2, \\ \mathfrak{Y}_3(t, \mathcal{S}_1, \mathcal{S}_2, \mathcal{I}_1, \mathcal{I}_2, \mathcal{T}, \mathcal{R}, \mathcal{E}) & = & \mu\beta\mathcal{S}_1 - (\sigma + \varsigma + \alpha)\mathcal{I}_1, \\ \mathfrak{Y}_4(t, \mathcal{S}_1, \mathcal{S}_2, \mathcal{I}_1, \mathcal{I}_2, \mathcal{T}, \mathcal{R}, \mathcal{E}) & = & \beta\mathcal{S}_2 - (\varrho + \varsigma + \alpha)\mathcal{I}_2, \\ \mathfrak{Y}_5(t, \mathcal{S}_1, \mathcal{S}_2, \mathcal{I}_1, \mathcal{I}_2, \mathcal{T}, \mathcal{R}, \mathcal{E}) & = & \varrho\mathcal{I}_2 - (\chi + \varsigma)\mathcal{T}, \\ \mathfrak{Y}_6(t, \mathcal{S}_1, \mathcal{S}_2, \mathcal{I}_1, \mathcal{I}_2, \mathcal{T}, \mathcal{R}, \mathcal{E}) & = & \sigma\mathcal{I}_1 + \chi\mathcal{T} - (\varsigma + \varpi)\mathcal{R}, \\ \mathfrak{Y}_7(t, \mathcal{S}_1, \mathcal{S}_2, \mathcal{I}_1, \mathcal{I}_2, \mathcal{T}, \mathcal{R}, \mathcal{E}) & = & (\mathcal{I}_1 + \mathcal{I}_2)\gamma - \kappa\mathcal{E}. \end{cases} \quad (16)$$

System (16) can also be written as

$$\begin{cases} {}^cD_{0^+}^{\Re}\mathfrak{Y}(t) = & \mathcal{X}(t, \mathfrak{Y}(t)), \ 0 \leq t \leq v, \ 0 < \Re \leq 1, \\ \mathfrak{Y}(0) = \mathfrak{Y}_0, \end{cases} \quad (17)$$

where

$$\begin{cases} \mathfrak{Y}(t) = & \mathcal{S}_1, \\ & \mathcal{S}_2, \\ & \mathcal{I}_1, \\ & \mathcal{I}_2, \\ & \mathcal{T}, \\ & \mathcal{R}, \\ & \mathcal{E}. \end{cases} \begin{cases} , \mathfrak{Y}_0(t) = & \mathcal{S}_{10}, \\ & \mathcal{S}_{20}, \\ & \mathcal{I}_{10}, \\ & \mathcal{I}_{20}, \\ & \mathcal{T}_0, \\ & \mathcal{R}_0, \\ & \mathcal{E}_0. \end{cases} \begin{cases} \mathcal{X}(t, \mathfrak{Y}(t)) = \mathfrak{Y}_1(t, \mathcal{S}_1, \mathcal{S}_2, \mathcal{I}_1, \mathcal{I}_2, \mathcal{T}, \mathcal{R}, \mathcal{E}) \\ \mathfrak{Y}_2(t, \mathcal{S}_1, \mathcal{S}_2, \mathcal{I}_1, \mathcal{I}_2, \mathcal{T}, \mathcal{R}, \mathcal{E}) \\ \mathfrak{Y}_3(t, \mathcal{S}_1, \mathcal{S}_2, \mathcal{I}_1, \mathcal{I}_2, \mathcal{T}, \mathcal{R}, \mathcal{E}) \\ \mathfrak{Y}_4(t, \mathcal{S}_1, \mathcal{S}_2, \mathcal{I}_1, \mathcal{I}_2, \mathcal{T}, \mathcal{R}, \mathcal{E}) \\ \mathfrak{Y}_5(t, \mathcal{S}_1, \mathcal{S}_2, \mathcal{I}_1, \mathcal{I}_2, \mathcal{T}, \mathcal{R}, \mathcal{E}) \\ \mathfrak{Y}_6(t, \mathcal{S}_1, \mathcal{S}_2, \mathcal{I}_1, \mathcal{I}_2, \mathcal{T}, \mathcal{R}, \mathcal{E}) \\ \mathfrak{Y}_7(t, \mathcal{S}_1, \mathcal{S}_2, \mathcal{I}_1, \mathcal{I}_2, \mathcal{T}, \mathcal{R}, \mathcal{E}) \end{cases} \quad (18)$$

Here, applying Lemma (2), we get the following

$$\mathfrak{Y}(t) = \mathfrak{Y}_o(t) + \frac{1}{\vartheta(\Re)} \int_0^t (t - \hbar)^{\Re-1} \mathcal{X}(\hbar, \mathfrak{Y}(\hbar)) d\hbar. \quad (19)$$

The following steps are important for further analysis of the system:
**(A)** Consider the constants $\mathfrak{P}_{\mathcal{X}}$, $\mathcal{J}_{\mathcal{X}}$, and $q \in [0, 1)$, in such a way that

$$|\mathcal{X}(t, \mathfrak{Y}(t))| \leq \mathfrak{P}_{\mathcal{X}}|\mathfrak{Y}|^q + \mathcal{J}_{\mathcal{X}}. \quad (20)$$

**(B)** Consider the constants $E\mathcal{X} > 0$, and $\mathfrak{Y}, \bar{\mathfrak{Y}} \in \mathcal{Z}$, in such a way that

$$|\mathcal{X}(t, \mathfrak{Y}) - \mathcal{X}(t, \bar{\mathfrak{Y}})| \leq E_{\mathcal{X}}[|\mathfrak{Y} - \bar{\mathfrak{Y}}|]. \quad (21)$$

The mapping of $U$ on $\mathcal{Z}$ is defined as

$$U\mathfrak{Y}(t) = \mathfrak{Y}_0(t) + \frac{1}{\vartheta(\Re)} \int_0^t (t - \hbar)^{\Re-1} \mathcal{X}(\hbar, \mathfrak{Y}(\hbar)) d\hbar. \quad (22)$$

At least one solution of (17) exists if the condition **A** and **B** satisfies. To obtain the solution for the proposed system this concept will be used in the upcoming theorem. The system (9) possess at least one solution if the conditions **A** and **B** satisfies. The well-known fixed point

theorem of Schaefer will be employed to establish the proof of the theorem. The result is proved as follows:

**Step 1**: To prove that $U$ is continuous. Considered that $\mathfrak{Y}_j$ is continuous for $j = 1, 2, \ldots, 7$, then $\mathcal{X}(t, \mathfrak{Y}(t))$ is continuous. Also assume $\mathfrak{Y}_\iota, \mathfrak{Y} \in \mathcal{Z}$ in such a way that $\mathfrak{Y}_\iota \to \mathfrak{Y}$, then $U\mathfrak{Y}_\iota \to U\mathfrak{Y}$. Now taking

$$
\begin{aligned}
||U\mathfrak{Y}_\iota - U\mathfrak{Y}|| &= \max_{t \in [0,v]} \left| \frac{1}{\vartheta(\Re)} \int_0^t (t - \hbar)^{\Re-1} \mathcal{X}_\iota(\hbar, \mathfrak{Y}_\iota(\hbar)) d\hbar - \frac{1}{\vartheta(\Re)} \int_0^t (t - \hbar)^{\Re-1} \mathcal{X}(\hbar, \mathfrak{Y}(\hbar)) d\hbar \right| \\
&\leq \max_{t \in [0,v]} \int_0^t \left| \frac{(t - \hbar)^{\Re-1}}{\vartheta(\Re)} \right| |\mathcal{X}_\iota(\hbar, \mathfrak{Y}_\iota(\hbar)) - \mathcal{X}(\hbar, \mathfrak{Y}(\hbar))| d\hbar \\
&\leq \frac{v^\Re E_\mathcal{X}}{\vartheta(\Re + 1)} ||\mathfrak{Y}_\iota - \mathfrak{Y}|| \to 0 \quad as \quad \iota \to \infty.
\end{aligned}
\tag{23}
$$

Since $\mathcal{X}$ is continuous, which follows that $U\mathfrak{Y}_\iota \to U\mathfrak{Y}$; thus we get that the operator $U$ is continuous.

**Step 2**: In this step the boundedness of the $U$ will be proved. Assume that $\mathfrak{Y} \in \mathcal{Z}$, then the operator $U$ holds the below stated results

$$
\begin{aligned}
||U\mathfrak{Y}|| &= \max_{t \in [0,v]} \left| \mathfrak{Y}_o(t) + \frac{1}{\vartheta(\Re)} \int_0^t (t - \hbar)^{\Re-1} \mathcal{X}(\hbar, \mathfrak{Y}(\hbar)) d\hbar \right| \\
&\leq |\mathfrak{Y}_0| \max_{t \in [0,v]} \frac{1}{\vartheta(\Re)} \int_0^t |(t - \hbar)^{\Re-1}| ||\mathcal{X}(\hbar, \mathfrak{Y}(\hbar))| d\hbar \\
&\leq |\mathfrak{Y}_0| + \frac{v^\Re}{\vartheta(\Re + 1)} [P_\mathcal{X} ||\mathfrak{Y}||^q + J_\mathcal{X}].
\end{aligned}
\tag{24}
$$

Now, we will prove the boundedness of $U(S)$, for a bounded subset $S$ of $\mathcal{Z}$. Assume that $\mathfrak{Y} \in S$, where $S$ is bounded, thus we get $P \geq 0$ such as

$$
||\mathfrak{Y}|| \leq P, \forall \mathfrak{Y} \in S.
\tag{25}
$$

For any $\mathfrak{Y} \in S$, we have

$$
||UW|| \leq |\mathfrak{Y}_0| + \frac{v^\Re}{\vartheta(\Re + 1)} [P_\mathcal{X} ||\mathfrak{Y}||^q + J_\mathcal{X}] \leq |\mathfrak{Y}_0| + \frac{v^\Re}{\vartheta(\Re + 1)} [P_\mathcal{X} P^q + J_\mathcal{X}].
\tag{26}
$$

Hence, $U(S)$ is bounded.

**Step 3**: To prove the equi-continuity, let $t_1, t_2 \in [0, v]$ such as $t_1 \geq t_2$ then we have

$$
\begin{aligned}
|U\mathfrak{Y}(t_1) - U\mathfrak{Y}(t_1)| &= \left| \frac{1}{\vartheta(\Re)} \int_0^{t_1} |(t_1 - \hbar)^{\Re-1}| ||\mathcal{X}(\hbar, \mathfrak{Y}(\hbar))| d\hbar - \frac{1}{\vartheta(\Re)} \int_0^{t_2} |(t_2 - \hbar)^{\Re-1}| ||\mathcal{X}(\hbar, \mathfrak{Y}(\hbar))| d\hbar \right| \\
&\leq \left| \frac{1}{\vartheta(\Re)} \int_0^{t_1} |(t_1 - \hbar)^{\Re-1}| - \frac{1}{\vartheta(\Re)} \int_0^{t_2} |(t_2 - \hbar)^{\Re-1}| \right| |\mathcal{X}(\hbar, \mathfrak{Y}(\hbar))| d\hbar \\
&\leq \frac{v^\Re}{\vartheta(\Re + 1)} [P_\mathcal{X} ||\mathfrak{Y}||^q + M_\mathcal{X}] [t_1^\Re - t_2^\Re] \to 0 \ as \ t_1 \to t_2.
\end{aligned}
\tag{27}
$$

By applying the Arzela-Ascoli theorem, we can achieve the relative compactness of $U(S)$.

**Step 4**: Let us consider a set as

$$\mathfrak{A} = \{\mathfrak{Y} \in \mathcal{Z} : \mathfrak{Y} = \wp U\mathfrak{Y}, \wp \in (0,1)\}. \tag{28}$$

To show the boundedness of $\mathfrak{A}$, let $\mathfrak{Y} \in \mathfrak{A}$, such as $t \in [0, v]$, then we have

$$||\mathfrak{Y}|| = \wp||U\mathfrak{Y}|| \leq \wp\left[|\mathfrak{Y}_0|\frac{v^\Re}{\vartheta(\Re+1)}[P_\mathcal{X}||\mathfrak{Y}||^q + M_\mathcal{X}]\right]. \tag{29}$$

Thus, we get that $\mathfrak{A}$ is bounded. Hence, through the theorem of Schaefer, the operator $U$ has a fixed point; this implies that the system (9) possesses at least one solution. The Theorem (4) holds for $\frac{v^\Re P_\mathcal{X}}{\vartheta(\Re+1)} < 1$, if the condition ($H1$) satisfies for $q = 1$. The system (17) possess a unique solution if $\frac{v^\Re P_\mathcal{X}}{\vartheta(\Re+1)} < 1$ holds. Assume that $\mathfrak{Y}, \bar{\mathfrak{Y}} \in \mathcal{Z}$ and using the Banach's contraction theorem, we get

$$\begin{aligned}||U\mathfrak{Y} - U\bar{\mathfrak{Y}}|| &\leq \max_{t\in[0,v]}\frac{1}{\vartheta(\Re)}\int_0^t |(t-\hbar)^{\Re-1}||\mathcal{X}(\hbar,\mathfrak{Y}(\hbar)) - \mathcal{X}(\hbar,\bar{\mathfrak{Y}}(\hbar))|d\hbar \\ &\leq \frac{v^\Re P_\mathcal{X}}{\vartheta(\Re+1)}||\mathfrak{Y} - \bar{\mathfrak{Y}}||.\end{aligned} \tag{30}$$

Therefore, there exists a fixed point of $U$. Thus, the system (17) has a unique solution.

## 5 Ulam-Hyers stability

The notion of stability was originally proposed in 1940 by Ulam and then further developed by Hyers [37, 38]. Currently, the theory of Ulam-Hyers stability is applied in various research areas of science. Below are some significant concepts to consider regarding this theory:

Suppose there is an operator $\mathcal{K} : \mathcal{Z} \to \mathcal{Z}$, such that

$$\mathcal{K}\mathfrak{Y} = \mathfrak{Y} \quad for \quad \mathfrak{Y} \in \mathcal{Z}. \tag{31}$$

If $\mathfrak{Y} \in \mathcal{Z}$ is any solution and $\ell > 0$, we have

$$||\mathfrak{Y} - \mathcal{K}\mathfrak{Y}|| \leq \ell \quad for \quad t \in [0, v], \tag{32}$$

then, the Eq (31) is Ulam-Hyers type stable (UHS).

A unique solution $\bar{\mathfrak{Y}}$ exists for Eq (31) with $C_q > 0$ and the below stated satisfies

$$||\bar{\mathfrak{Y}} - \mathfrak{Y}|| \leq C_q\ell, \quad where \quad 0 \leq t \leq v. \tag{33}$$

The Eq (31) will be generalized UHS, take any solution $\mathfrak{Y}$ of Eq (31) and $\bar{\mathfrak{Y}}$ be any other solution of (31), then

$$||\bar{\mathfrak{Y}} - \mathfrak{Y}|| \leq \mathcal{B}(\ell), \tag{34}$$

where 0 is mapping of 0 and $\mathcal{B} \in C(\mathbb{R}, \mathbb{R})$. If the following conditions holds true, then the solution $\bar{\mathfrak{Y}} \in \mathcal{Z}$ satisfies (33)

(1) $|\Gamma| \leq \ell$, in which $0 \leq t \leq v$, $\quad \Gamma \in C([0, v]; \mathbb{R})$.
(2) $\mathcal{K}\bar{\mathfrak{Y}}(T) = \bar{\mathfrak{Y}} + \Gamma(T), \quad in \ which \quad 0 \leq t \leq v.$

After small perturbation, system (17) implies

$$
\begin{cases}
{}^{c}D_{0^+}^{\Re}\mathfrak{Y}(t) = & \mathcal{X}(t, \mathfrak{Y}(t)) + \Gamma(t), \\
\mathfrak{Y}(0) = \mathfrak{Y}_0.
\end{cases}
\tag{35}
$$

The system (35) fulfills the following

$$
|\mathfrak{Y}(t) - U\mathfrak{Y}(t)| \leq b\ell, \quad in \ which \quad b = \frac{v^{\Re}}{\vartheta(\Re+1)}.
\tag{36}
$$

Utilizing Remark (5) and Lemma (2), we can easily prove this result. The solution of (17) will be UH-stable and GUH-stable on Lemma (5) if $\frac{v^{\Re}E_{\mathcal{X}}}{\vartheta(\Re+1)} < 1$ satisfies. For the proof, take any solution $\mathfrak{Y} \in \mathcal{Z}$ of the system (35) and a unique solution $\bar{\mathfrak{Y}} \in \mathcal{Z}$ of (17), then we have

$$
\begin{aligned}
|\mathfrak{Y}(t) - \bar{\mathfrak{Y}}(t)| &= |\mathfrak{Y}(t) - \bar{\mathfrak{Y}}(t)| \\
&\leq |\mathfrak{Y}(t) - U\bar{\mathfrak{Y}}(t)| \\
&\leq |\mathfrak{Y}(t) - U\bar{\mathfrak{Y}}(t)| \\
&\leq b\ell + \frac{v^{\Re}E_{\mathcal{X}}}{\vartheta(\Re+1)}|\mathfrak{Y}(t) - \bar{\mathfrak{Y}}(t)| \\
&\leq \frac{b\ell}{1 - \frac{v^{\Re}E_{\mathcal{X}}}{\vartheta(\Re+1)}},
\end{aligned}
\tag{37}
$$

which shows that the system (16) is GUH-stable and UH-stable. Assume that $\Xi \in C[[0, v], \mathbb{R}]$, then (31) is Ulam-Hyers-Rassias (UHR) stable if $\mathfrak{Y} \in \mathcal{Z}$ is any solution and

$$
||\mathfrak{Y} - \mathcal{K}\mathcal{J}|| \leq \Xi(t)\ell, \ for \ t \in [0, v] \ and \ \ell > 0.
\tag{38}
$$

we can get a unique solution $\bar{\mathfrak{Y}}$ for the system (31) such that $C_q > 0$ satisfying

$$
||\bar{\mathfrak{Y}} - \mathfrak{Y}|| \leq C_q\Xi(t)\ell, \ \forall \ t \in [0, v].
\tag{39}
$$

Consider any solution $\mathfrak{Y}$ of (38) and a unique solution $\bar{\mathfrak{Y}}$ such that

$$
||\bar{\mathfrak{Y}} - \mathfrak{Y}|| \leq C_{q,\Xi}\Xi(t)\ell, \quad \forall \quad t \in [0, v].
\tag{40}
$$

where $\ell > 0$ and $\Xi \in C[[0, v], \mathbb{R}]$ such as $C_{q,\Xi}$. Then, the Eq (31) is generalized UHR-stable. If the following conditions holds, then the solution $\bar{\mathfrak{Y}} \in \mathcal{Z}$ satisfies (33)

(a) $|\Gamma(t)| \leq \ell\Xi(t), \ in \ which \quad 0 \leq t \leq v, \quad \Gamma(t) \in C([0, v]; \mathbb{R}).$

(b) $\mathcal{K}\bar{\mathfrak{Y}}(t) = \bar{\mathfrak{Y}} + \Gamma(t), \quad in \ which \quad 0 \leq t \leq v.$ For the perturb system (5), the following exists

$$
|\mathfrak{Y}(t) - U\mathfrak{Y}(t)| \leq b\Xi(t)\ell, \ where \quad b = \frac{v^{\Re}}{\vartheta(\Re+1)}.
\tag{41}
$$

**Proof**. By utilizing Remark (5) and Lemma (2), the result can be easily obtained. The solution of system (17) will be GUHR-stable and UHR-stable on Lemma (5), if $\frac{v^{\Re}L_{\mathcal{X}}}{\vartheta(\Re+1)} < 1$.

**Proof**. To get the required result, assume that $\mathfrak{Y} \in \mathcal{Z}$ be a solution and $\bar{\mathfrak{Y}} \in \mathcal{Z}$ be an unique solution of the system (17), then

$$
\begin{aligned}
|\mathfrak{Y}(t) - \bar{\mathfrak{Y}}(t)| &= |\mathfrak{Y}(t) - \bar{\mathfrak{Y}}(t)| \\
&\leq |\mathfrak{Y}(t) - U\bar{\mathfrak{Y}}(t)| \\
&\leq |\mathfrak{Y}(t) - U\bar{\mathfrak{Y}}(t)| \\
&\leq b\Xi(t)\ell + \frac{v^{\Re}E_{\mathcal{X}}}{\vartheta(\Re+1)}|\mathfrak{Y}(t) - \bar{\mathfrak{Y}}(t)| \\
&\leq \frac{b\Xi(t)\ell}{1 - \dfrac{v^{\Re}E_{\mathcal{X}}}{\vartheta(\Re+1)}}.
\end{aligned}
\tag{42}
$$

Therefore the Eq (17) solution is GUHR and UHR-stable.

## 6 Iterative method for solution

Here, we will present a numerical method for elucidating the dynamic behavior of the system (9) of cryptosporidiosis disease. The main approach methodology, we utilized is as follows

$$
{}_0^C D_t^{\Re} y(t) = q(t, y(t)).
\tag{43}
$$

The utilization of the fundamental theorem of fractional-calculus on Eq (43) can yield the following outcome;

$$
y(t) - y(0) = \frac{1}{\vartheta(\Re)} \int_0^t q(\varkappa, y(\varkappa))(t - \varkappa)^{\Re-1} d\varkappa,
\tag{44}
$$

at $t = t_{\theta+1}$, $\theta = 0, 1, \ldots$, we have

$$
y(t_{\theta+1}) - y(0) = \frac{1}{\vartheta(\Re)} \int_0^{t_{\theta+1}} (t_{\theta+1} - t)^{\Re-1} q(t, y(t)) dt,
\tag{45}
$$

and

$$
y(t_{\theta}) - y(0) = \frac{1}{\vartheta(\Re)} \int_0^{t_{\theta}} (t_{\theta} - t)^{\Re-1} q(t, y(t)) dt.
\tag{46}
$$

The Eqs (45) and (46) yields that

$$
\begin{aligned}
y(t_{\theta+1}) = \ \ & y(t_{\theta}) + \underbrace{\frac{1}{\vartheta(\Re)} \int_0^{t_{\theta+1}} (t_{\theta+1} - t)^{\Re-1} q(t, y(t)) dt}_{\mathcal{H}_{\Re,1}} \\
& - \underbrace{\frac{1}{\vartheta(\Re)} \int_0^{t_{\theta}} (t_{\theta} - t)^{\Re-1} q(t, y(t)) dt}_{\mathcal{H}_{\Re,2}}.
\end{aligned}
\tag{47}
$$

where

$$
\mathcal{H}_{\Re,1} = \frac{1}{\vartheta(\Re)} \int_0^{t_{\theta+1}} (t_{\theta+1} - t)^{\Re-1} q(t, y(t)) dt,
\tag{48}
$$

and

$$\mathcal{H}_{\Re,2} = \frac{1}{\vartheta(\Re)} \int_0^{t_\theta} (t_\theta - t)^{\Re-1} q(t, y(t)) dt. \tag{49}$$

Approximating $q(t, y(t))$ by utilizing lagrange approximation, we obtain

$$
\begin{aligned}
\mathcal{Q}(t) &\simeq \frac{t - t_{\theta-1}}{t_\theta - t_{\theta-1}} q(t_\theta, y_\theta) + \frac{t - t_\theta}{t_{\theta-1} - t_\theta} q(t_{\theta-1}, y_{\theta-1}) \\
&= \frac{q(t_\theta, y_\theta)}{w}(t - t_{\theta-1}) - \frac{q(t_{\theta-1}, y_{\theta-1})}{w}(t - t_\theta).
\end{aligned}
\tag{50}
$$

Moreover, we have

$$
\begin{aligned}
\mathcal{H}_{\Re,1} &= \frac{q(t_\theta, y_\theta)}{w\vartheta(\Re)} \int_0^{t_{\theta+1}} (t_{\theta+1} - t)^{\Re-1}(t - t_{\theta-1}) dt \\
&\quad - \frac{q(t_{\theta-1}, y_{\theta-1})}{w\vartheta(\Re)} \int_0^{t_{\theta+1}} (t_{\theta+1} - t)^{\Re-1}(t - t_\theta) dt.
\end{aligned}
\tag{51}
$$

$$
\begin{aligned}
\mathcal{H}_{\Re,1} &= \frac{q(t_\theta, y_\theta)}{w\vartheta(\Re)} \left[ \frac{2w}{\Re} t_{\theta+1}^\Re - \frac{t_{\theta+1}^{\Re+1}}{\Re+1} \right] \\
&\quad - \frac{q(t_{\theta-1}, y_{\theta-1})}{w\vartheta(\Re)} \left[ \frac{w}{\Re} t_{\theta+1}^\Re - \frac{1}{\Re+1} t_{\theta+1}^{\Re+1} \right].
\end{aligned}
\tag{52}
$$

Similarly, we obtain

$$\mathcal{H}_{\Re,2} = \frac{1}{\vartheta(\Re)} \int_0^{t_\theta} (t_\theta - t)^{\Re-1} \left[ \frac{q(t_\theta, y_\theta)}{w}(t - t_{\theta-1}) - \frac{q(t_{\theta-1}, y_{\theta-1})}{w}(t - t_\theta) \right] dt. \tag{53}$$

After further evaluation we get that

$$
\begin{aligned}
\mathcal{H}_{\Re,2} &= \frac{q(t_\theta, y_\theta)}{w\vartheta(\Re)} \left[ \frac{w}{\Re} t_\theta^\Re - \frac{t_\theta^{\Re+1}}{\Re+1} \right] \\
&\quad + \frac{q(t_{\theta-1}, y_{\theta-1})}{w\vartheta(\Re)} \left[ \frac{1}{\Re+1} t_\theta^{\Re+1} \right].
\end{aligned}
\tag{54}
$$

By substituting (53) and (54) in (47) we get approximate solution for the system (43), such as

$$
\begin{aligned}
y(t_{\theta+1}) &= y(t_\theta) + \frac{q(t_\theta, y_\theta)}{w\vartheta(\Re)} \left[ \frac{2w t_{\theta+1}^\Re}{\Re} - \frac{t_{\theta+1}^{\Re+1}}{\Re+1} + \frac{w}{\Re} t_\theta^\Re - \frac{t_{\theta+1}^{\Re+1}}{\Re+1} \right] \\
&\quad + \frac{q(t_{\theta-1}, y_{\theta-1})}{w\vartheta(\Re)} \left[ -\frac{w}{\Re} t_{\theta+1}^\Re + \frac{t_{\theta+1}^{\Re+1}}{\Re+1} + \frac{t_\theta^{\Re+1}}{\Re+1} \right].
\end{aligned}
\tag{55}
$$

Now, we will utilize the above numerical scheme to graphically represent the solution pathways of the model of cryptosporidiosis infection. We will conduct various numerical scenarios to demonstrate how input factors impact the dynamics of the infection. Based on our results, we will suggest effective control policies to reduce the frequency of infection in the population. Through simulations, we present a time series analysis of strong immunity infected, weak immunity infected, treated individuals, and recovered individuals. For numerical reasons, the values of the state variables and input parameters are supposed.

The role of fractional order on the infection level has been conceptualized in the first simulation illustrated in Figs 1 and 2. The solution pathways of infected, treated, and recovered individuals are presented with variations of the fractional order $\Re$. It has been observed that the fractional order has a significant impact on reducing the level of infection. Therefore, public health officials should consider manipulating this parameter as an effective approach for managing and controlling infection in society. In Fig 3, the impact of transmission probability $\varpi$ on the time series of the recommended system has been visualized. At the same time, the effect of the input parameter $\tau$ has been conceptualized in Fig 4. These parameters play a

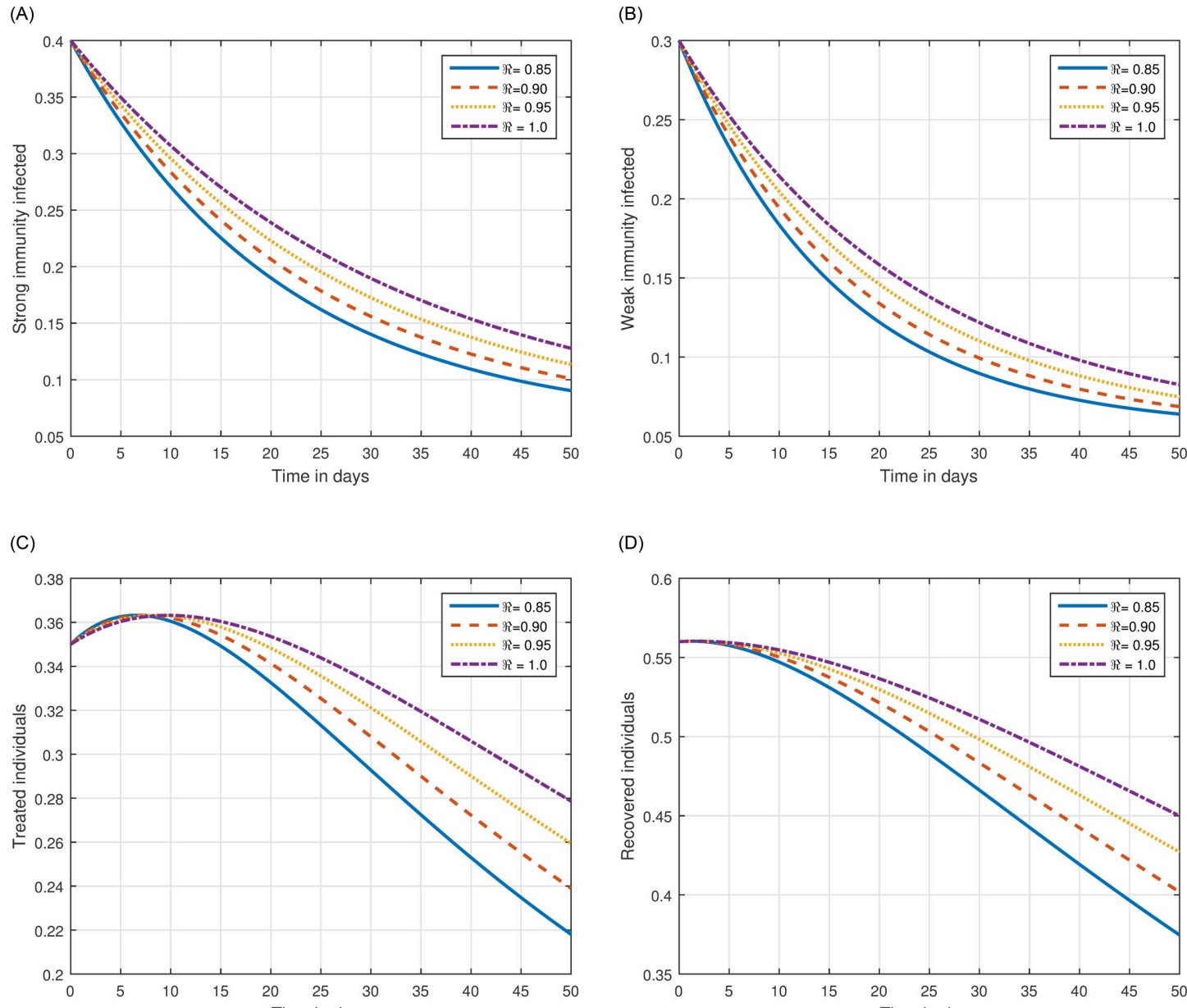

**Fig 1. Numerical visualization of the time series of the system (9) of cryptosporidiosis with different values of fractional order $\Re$, i.e., $\Re$ = 0.85, 0.90, 0.95, 1.0.**

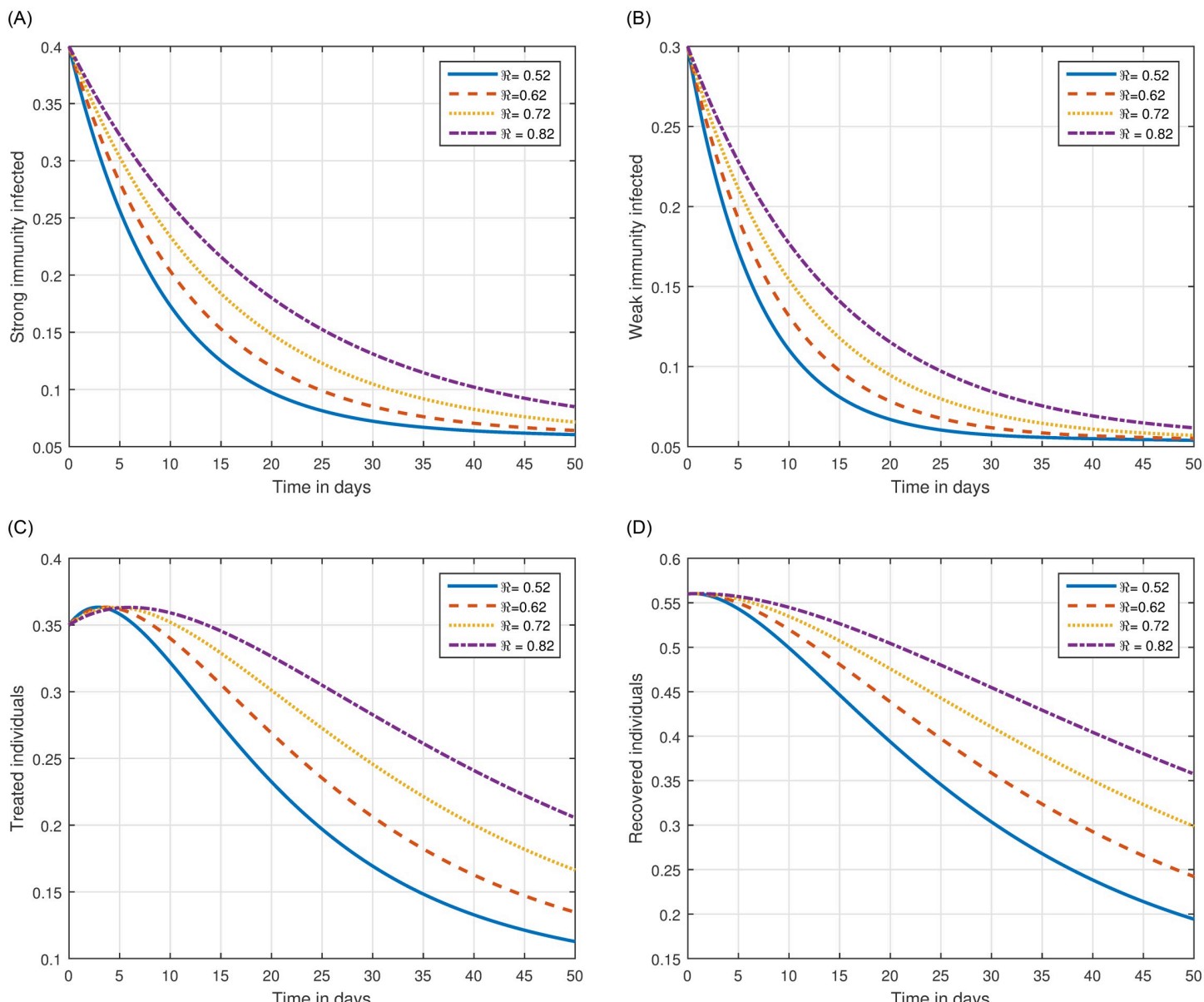

**Fig 2. Numerical visualization of the dynamical behaviour of the system (9) of cryptosporidiosis with different values of fractional order ℜ, i.e., ℜ = 0.52, 0.62, 0.72, 0.82.**

critical role and can increase the risk of infection in the community. We demonstrated the significant contribution of the treatment factor in effectively controlling and preventing cryptosporidiosis parasitic infection in Fig 5. We propose that utilizing treatment as a control parameter can effectively reduce the level of infection in the population.

In the last simulation presented in Fig 6, we have shown the impact of the loss rate of immunity on the transmission dynamics of cryptosporidiosis. We noticed that this input factor makes the control of the infection more difficult and destabilizes the dynamics of the infection. We have shown that treatment rate and the fractional order are attractive parameters for the

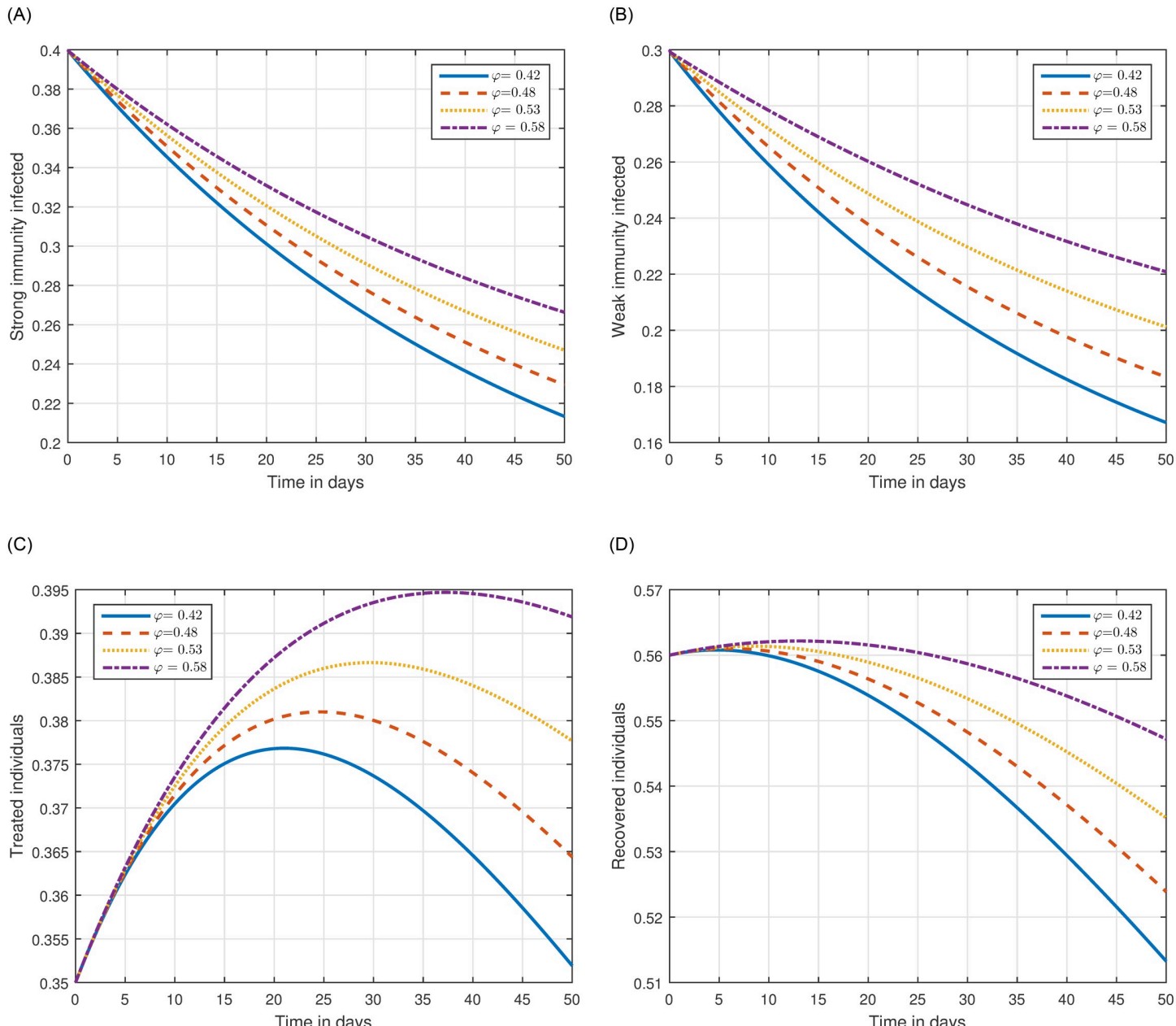

**Fig 3. Plotting the infected individuals of the suggested system (9) of cryptosporidiosis with different values of the input parameter $\varphi$, i.e., $\varphi$ = 0.42, 0.48, 0.53, 0.58.**

control of the infection. Therefore, we suggested that treatment through medication and an index of memory can be used to control the level of infection in society.

## 7 Concluding

In this research, we formulated the dynamics of the parasitic infection cryptosporidiosis with strong and weak immunity through Caputo fractional derivative. We presented the basic theory of fractional calculus to examine our model. The investigation concentrates on the

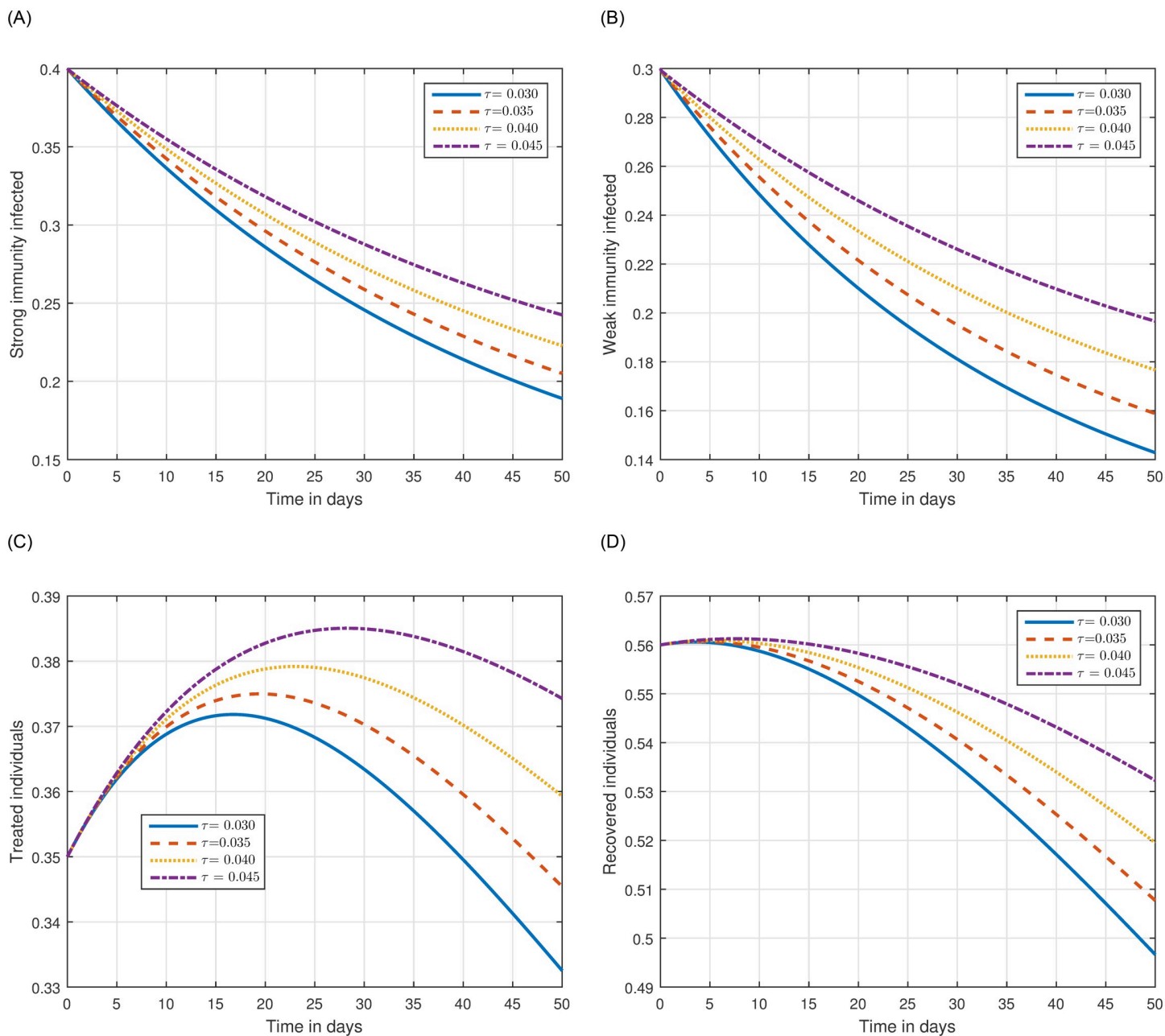

**Fig 4. Plotting the infected individuals of system (9) of cryptosporidiosis with different values of the input parameter τ, i.e., τ = 0.030, 0.035, 0.040, 0.045.**

dynamical behavior and qualitative analysis of cryptosporidiosis infection. We investigated the model for steady states and determined the threshold parameter with the help of the Next-generation matrix method. The dynamics of the infection are analyzed using the fixed-point theorem within the Banach and Schaefer framework to determine the uniqueness and existence of the solution. The Ulam-Hyers stability of the cryptosporidiosis infection system was established under appropriate conditions. We demonstrate a numerical approach to investigate the tracking path behavior of the model and forecast the effect of different

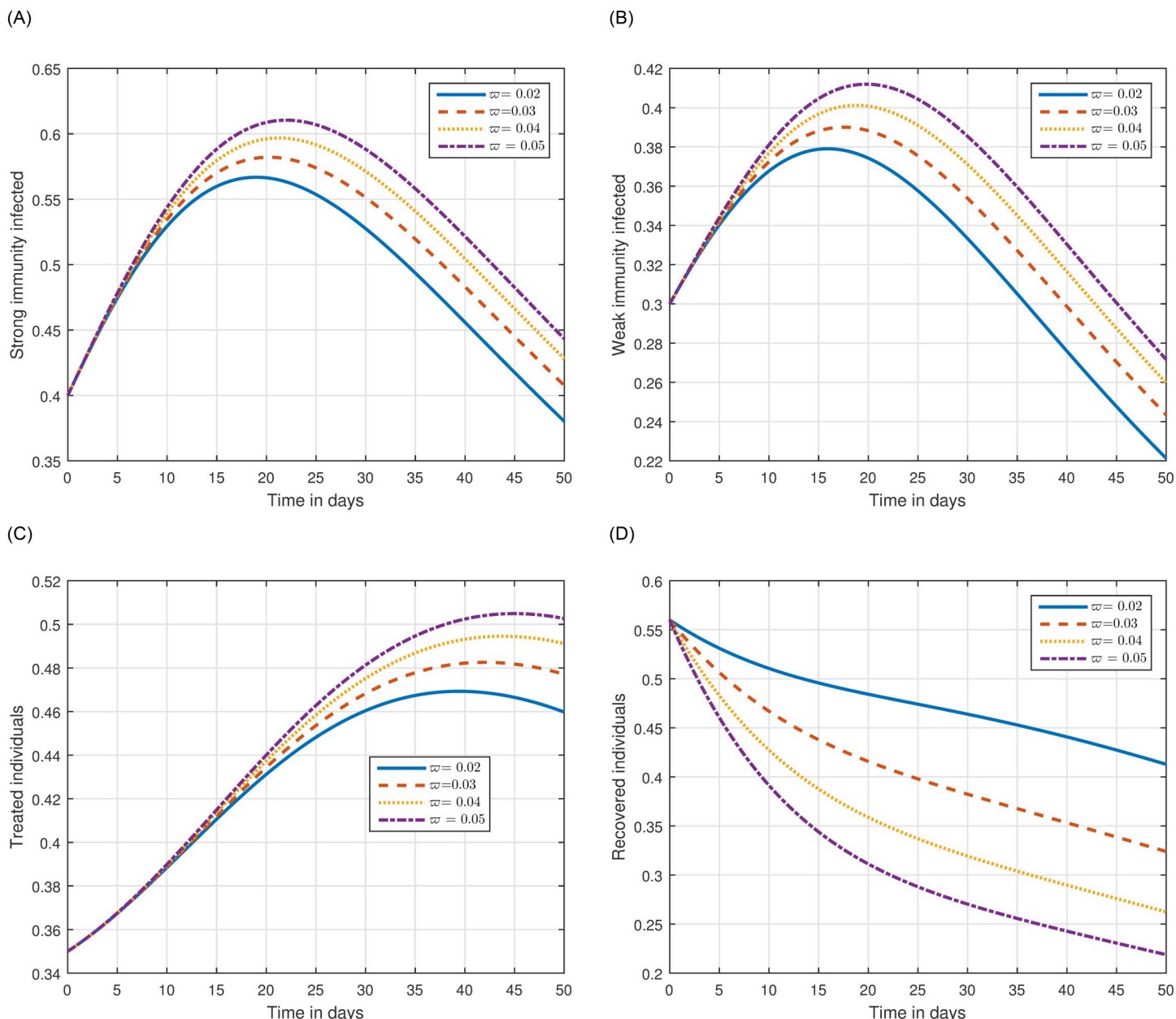

**Fig 5. Graphical view analysis of the time series of the model (9) of cryptosporidiosis infection with different values of the input parameter $\varphi$, i.e., $\varphi = 0.02, 0.03, 0.04, 0.05$.**

parameters on the infection dynamics. Through numerical analysis, we have demonstrated how various input factors affect the system's output and identified the most crucial factors in the system. Our analysis offered distinct perspectives on the infection dynamics to the policymakers. Cryptosporidiosis exhibits seasonal patterns, with higher incidence rates in the summer months. In our future work, we will include seasonal effects to better capture the transmission behavior.

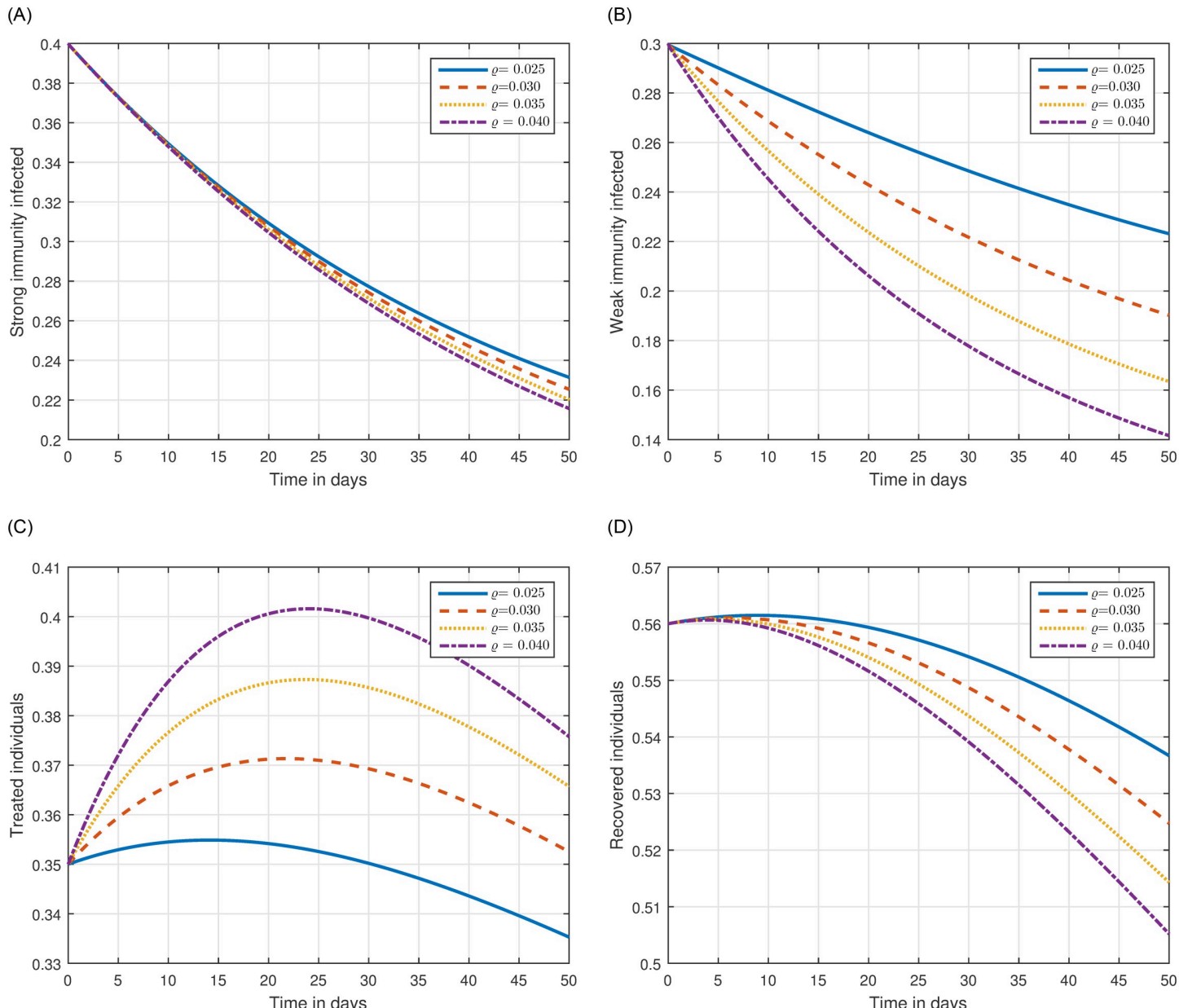

**Fig 6. Illustration of the time series of the system (9) of cryptosporidiosis with various values of the input parameter $\varrho$, i.e., $\varrho$ = = 0.025, 0.030, 0.035, 0.040.**

## Author Contributions

**Conceptualization:** Tao-Qian Tang, Zahir Shah, Asif Jan.

**Data curation:** Tao-Qian Tang, Zahir Shah, Ciprian Tanasescu, Asif Jan.

**Formal analysis:** Tao-Qian Tang.

**Funding acquisition:** Narcisa Vrinceanu.

**Investigation:** Rashid Jan, Zahir Shah, Ciprian Tanasescu.

**Methodology:** Tao-Qian Tang, Rashid Jan, Zahir Shah, Asif Jan.

**Project administration:** Narcisa Vrinceanu.

**Resources:** Narcisa Vrinceanu, Ciprian Tanasescu.

**Software:** Rashid Jan, Asif Jan.

**Supervision:** Zahir Shah.

**Validation:** Rashid Jan, Narcisa Vrinceanu, Ciprian Tanasescu.

**Visualization:** Narcisa Vrinceanu.

**Writing – original draft:** Tao-Qian Tang, Asif Jan.

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
