## [Decision Letter · Decision Letter 0]

15 Aug 2023

PONE-D-23-15107A fractional perspective on the transmission dynamics of a parasitic infection, considering the impact of both strong and weak immunityPLOS ONE

Dear Dr. Shah,

Thank you for submitting your manuscript to PLOS ONE. After careful consideration, we feel that it has merit but does not fully meet PLOS ONE’s publication criteria as it currently stands. Therefore, we invite you to submit a revised version of the manuscript that addresses the points raised during the review process.

We look forward to receiving your revised manuscript.

Kind regards,

Maria Stefania Latrofa

Academic Editor

PLOS ONE

Journal Requirements:

Reviewers' comments:

Reviewer's Responses to Questions

**Comments to the Author**

1. Is the manuscript technically sound, and do the data support the conclusions?

Reviewer #1: Partly

2. Has the statistical analysis been performed appropriately and rigorously? 

Reviewer #1: I Don't Know

3. Have the authors made all data underlying the findings in their manuscript fully available?

Reviewer #1: Yes

4. Is the manuscript presented in an intelligible fashion and written in standard English?

Reviewer #1: Yes

5. Review Comments to the Author

Reviewer #1: The authors of this manuscript study cryptosporidiosis, an infection caused by a parasite found in stool. Creating a mathematical model describing the spread of this infection is an interesting research issue. The introduction affords additional information on the importance of creating a mathematical model that may help to find a way of controlling the spreading of this infection.

Section 3 is devoted to building up a set of 7 coupled equations, called (8), which, according to the authors should contribute to control the spreading of this infection. The authors after creating the model (8) makes an abrupt jump to the model (9) that is identical to the model (8). The only difference is that the ordinary time derivative is replaced by the Caputo fractional derivatives with the index R. When R = 1, the model (9) becomes identical to the model (8).

The authors claim that the model (9) is better than model (8). Why? The readers would be convinced that this is the right direction to follow if the authors made a comparison with real experimental data and showed that (9) reproduces the experimental results more accurately.

With no comparison of this kind, it is difficult to find good reasons for the publication of this manuscript.

Would the authors be able to explain how their model can contribute to the control of this infection without a comparison between theory and experiment?

At page 23 the authors write: “The solution pathway of infected, treated and recover individuals are presented with variation of the fractional parameter R. We observed that the fractional order parameter has a significant impact in reducing the level of infection. Hence, controlling this parameter can be an effective strategy to regulate the level of infection.” This is an interesting property, but if a comparison with real data shows a very good agreement with R = 0.85, realizing a significant reduction of infection, which kind of procedures

the health organization should use to produce results corresponding to R = 0.85?

I think that the authors should answer this important question. If the mathematical discussion to get the solution of the set of Eq. (9) affords the answer to this question, the authors should help the readers to understand it.

6. PLOS authors have the option to publish the peer review history of their article (what does this mean?). If published, this will include your full peer review and any attached files.

Reviewer #1: No

---

## [Author Response · Author response to Decision Letter 0]

14 Oct 2023

Response Letters

Title: A fractional perspective on the transmission dynamics of a parasitic infection, considering the impact of both strong and weak immunity

PONE-D-23-15107

Dear Editor,

Please find attached our revised manuscript of the above paper. We are very grateful to the Editor for valuable comments and suggestions for the improvement of our work. Reviewer #1: The authors of this manuscript study cryptosporidiosis, an infection caused by a parasite found in stool. Creating a mathematical model describing the spread of this infection is an interesting research issue. The introduction affords additional information on the importance of creating a mathematical model that may help to find a way of controlling the spreading of this infection. Section 3 is devoted to building up a set of 7 coupled equations, called (8), which, according to the authors should contribute to control the spreading of this infection. The authors after creating the model (8) makes an abrupt jump to the model (9) that is identical to the model (8). The only difference is that the ordinary time derivative is replaced by the Caputo fractional derivatives with the index R. When R = 1, the model (9) becomes identical to the model (8). Response: Thank you so much for your valuable comments and suggestions for the improvement of our work. Yes, we present our model in the framework of ordinary differential equations as well as fractional differential equations at the beginning. The first thing is that we can make a comparison between these two frameworks. Secondly, if memory is involved in the transmission phenomena of an infection, then this parameter may be used as a control parameter. Thirdly, moving to the fractional model from the classical ODE model can also provide a free variable for data analyses and with this variable, the researchers can fit real data more accurately than the classical model. Due to the advantages of the fractional operator, we choose to investigate the model in a fractional framework.

The authors claim that model (9) is better than model (8). Why? The readers would be convinced that this is the right direction to follow if the authors made a comparison with real experimental data and showed that (9) reproduces the experimental results more accurately. With no comparison of this kind, it is difficult to find good reasons for the publication of this manuscript. Response: Fractional epidemic models can offer advantages over classical epidemic models in certain situations due to their ability to capture more complex and realistic behaviors in the spread of infectious diseases. Fractional epidemic models are better than ordinary models because they possess memory effects, nonlocal effects, and hereditary properties that are

involved in complex biological phenomena. These models can also provide accurate information about the long--term behavior of different infections.

Would the authors be able to explain how their model can contribute to the control of this infection without a comparison between theory and experiment? Response: We comprehend the transmission dynamics of the infection and identified the key factors of the system. We have shown that treatment rate and index of memory are attractive parameter for the control of the infection. Therefore, we suggested that treatment through medication and index of memory can be used as control the infection in the society.

At page 23 the authors write: “The solution pathway of infected, treated and recover individuals are presented with variation of the fractional parameter R. We observed that the fractional order parameter has a significant impact in reducing the level of infection. Hence, controlling this parameter can be an effective strategy to regulate the level of infection.” This is an interesting property, but if a comparison with real data shows a very good agreement with R = 0.85, realizing a significant reduction of infection, which kind of procedures the health organization should use to produce results corresponding to R = 0.85? Response: Thank you so much for your nice comments on our work. The policymakers and health officials will check the involvement of index of memory or fractional factor in the transmission phenomena of the infection and then will further investigate how to reduce this parameter. We have shown that this is a promising parameter and can contribute to reduce the level of infection. Also, on the availability of real data, the fractional system has an extra parameter which makes the system more flexible to real data.

Thank you for all the work you have put into our manuscript, and please pass on our sincere thanks to the anonymous referees for their helpful comments and the Editor for his/her constructive suggestions.

Sincerely yours

---

## [Editor Report · Decision Letter 1]

16 Jan 2024

A fractional perspective on the transmission dynamics of a parasitic infection, considering the impact of both strong and weak immunity

PONE-D-23-15107R1

Dear Dr. Shah,

We’re pleased to inform you that your manuscript has been judged scientifically suitable for publication and will be formally accepted for publication once it meets all outstanding technical requirements.

Kind regards,

Maria Stefania Latrofa

Academic Editor

PLOS ONE

---

## [Editor Report · Acceptance letter]

20 Mar 2024

PONE-D-23-15107R1 

PLOS ONE

Dear Dr. Shah, 

I'm pleased to inform you that your manuscript has been deemed suitable for publication in PLOS ONE. Congratulations! Your manuscript is now being handed over to our production team.

Kind regards, 

on behalf of

Dr. Maria Stefania Latrofa 

Academic Editor

PLOS ONE